



# Validation and Error Estimation of AIRS MUSES CO Profiles with HIPPO, ATom and NOAA GML Aircraft Observations

Jennifer D.  Hegarty[1], Karen E. Cady-Pereira[1], Vivienne H. Payne[2], Susan S. Kulawik[3], John R. Worden[2], Valentin Kantchev[2,4], Helen M. Worden[5], Kathryn McKain[6,7], Jasna V. Pittman[8,9], Róisín Commane[10], Bruce C. Daube Jr.[8,9], and Eric A. Kort[11]

[1] Atmospheric and Environmental Research Inc., Lexington, Massachusetts, USA
[2] Jet Propulsion Laboratory, California Institute of Technology, Pasadena, CA, USA
[3] BAER Institute, 625 2nd Street, Suite 209, Petaluma, CA, USA
[4] Instrument Software and Science Data Systems, Pasadena, CA, USA
[5] Atmospheric Chemistry Observations and Modeling Laboratory, National Center for Atmospheric Research, Boulder, CO, USA
[6] Earth System Research Laboratory, National Oceanic and Atmospheric Administration, Boulder, CO, USA
[7] Cooperative Institute for Research in Environmental Sciences, University of Colorado, Boulder, CO, USA
[8] School of Engineering and Applied Sciences, Harvard University, Cambridge, MA, USA
[9] Department of Earth and Planetary Sciences, Harvard University, Cambridge, MA, USA
[10] Department of Earth and Environmental Science, Lamont-Doherty Earth Observatory, Columbia University, New York, NY, USA
[11] Climate and Space Sciences and Engineering, University of Michigan, Ann Arbor, MI, USA

*Correspondence:* Jennifer D. Hegarty (jhegarty@aer.com)

**Abstract.** Single footprint retrievals of carbon monoxide from the Atmospheric Infrared Sounder (AIRS) are evaluated using aircraft in situ observations. The aircraft data are from the HIAPER Pole-to-Pole (HIPPO, 2009–2011), the first three Atmospheric Tomography Mission (ATom, 2016–2017) campaigns and the National Oceanic and Atmospheric Administration (NOAA) Global Monitoring Laboratory (GML) Global Greenhouse Gas Reference Network Aircraft Program between 2006 - 2017.  The retrievals are obtained using an optimal estimation approach within the MUlti-SpEctra, MUlti-SpEcies, MUlti-Sensors (MUSES) algorithm. Retrieval biases and estimated errors are evaluated across a range of latitudes from the sub-polar to tropical regions over both ocean and land points.

AIRS MUSES CO profiles were compared with HIPPO, ATom, and NOAA GML aircraft observations with a coincidence of 9 hours and 50 km to estimate retrieval biases and standard deviations.  Comparisons were done for different pressure levels and column averages, latitudes, day, night, land, and ocean observations.  We find mean biases of + 6.6% +/- 4.6%, +0.6% +/- 3.2%, -6.1% +/- 3.0%, and 1.4% +/- 3.6%, for 750 hPa, 510 hPa, 287 hPa, and the column averages, respectively.  The mean standard deviation is 15%, 11%, 12%, and 9% at these same pressure





levels, respectively. Observation errors (theoretical errors) from the retrievals were found to be
broadly consistent in magnitude with those estimated empirically from ensembles of satellite
aircraft comparisons. The GML Aircraft Program comparisons generally had higher standard
deviations and biases than the HIPPO and ATom comparisons. Since the GML aircraft flights do
not go as high as the HIPPO and ATom flights, results from these GML comparisons are more

sensitive to the choice of method for extrapolation of the aircraft profile above the uppermost
measurement altitude.  The AIRS retrieval performance shows little sensitivity to surface type
(land or ocean) or day or night but some sensitivity to latitude. Comparisons to the NOAA GML
set spanning the years 2006–2017 show that the AIRS retrievals are able to capture the distinct
seasonal cycles but show a high bias of ~ 20% in the lower troposphere during the summer when

observed CO mixing ratios are at annual minimum values.  The retrieval bias drift was examined
over the same period and found to be small at < 0.5% over the 2006–2017 time period.

## 1.  Introduction

Carbon monoxide (CO) is produced by the combustion of fossil fuels and biofuels, wildfires and
agricultural biomass burning, and hydrocarbon oxidation.  It is a precursor to tropospheric ozone
and carbon dioxide and thus plays an important role in both atmospheric pollution and climate.
CO is removed from the atmosphere mainly through reactions with the hydroxyl radical (OH) and
influences the removal rates of other atmospheric pollutants.  CO has a chemical lifetime greater

than a week in the troposphere, which allows it to be transported long distances.  At the same time
the lifetime is short enough that concentrations generally remain spatially inhomogeneous.  It is
therefore a good tracer species whose uneven distribution can be used to analyze regional to global
transport processes from pollution sources (e.g., Edwards et al., 2004, 2006; Hegarty et al., 2009,
2010; Petetin et al., 2018; Panagi et al., 2020).

The satellite record of nadir CO observations began in 2000 with the Measurement of Pollution in
the Troposphere (MOPITT) instrument on the NASA Terra satellite (Drummond et al., 2010). The
nadir satellite CO record now includes datasets from the Atmospheric Infrared Spectrometer
(AIRS) on Aqua launched in 2002, The Scanning Imaging Absorption Spectrometer for
Atmospheric Chartography (SCIAMACHY) on Envisat launched in 2003, the Tropospheric

Emission Spectrometer (TES) on Aura launched in 2004, the Infrared Atmospheric Sounding
Interferometer (IASI) on the MetOp series beginning in 2006, the Cross-track Infrared Sounder



(CrIS) on Suomi-NPP launched in 2011, and most recently the Joint Polar Satellite System series and TROPOMI on the Sentinel-5 precursor in 2017. Satellite CO datasets have been used extensively in emission source attribution studies (e.g., Kopacz et al., 2010; Jiang et al., 2017) and

trend analyses (e.g., Worden et al., 2013a; Buchholz et al., 2021). Among the satellite instruments currently observing CO, AIRS and MOPITT have the longest continuous records making them the most suitable for trend analysis. Though the MOPITT data record begins two years earlier, AIRS has the advantage of a swath width approximately twice as large as MOPITT's enabling near global coverage in about a day as compared to about three days for MOPITT (Yurganov et al.,

80    2008).

Characterization of uncertainties is key for the effective use of any measurement in emission source attribution and trend studies. Ideally, the characterization of uncertainties in satellite datasets should include both quantification of biases and the validation of the error estimates associated with the remotely sensed products (von Clarmann et al., 2020). In this paper we present

an evaluation of these uncertainties for a new set of CO retrievals from AIRS. These retrievals differ from previous AIRS products in that they are derived from single footprint L1B radiances, rather than from radiances obtained from applying a cloud clearing algorithm to sets of nine footprints. Therefore, the spatial resolution of this new product is the native spatial resolution of the Level 1B radiances (15 km at nadir). The algorithm utilized here is the MUlti-SpEctra, MUlti-

SpEcies, MUlti-Sensors (MUSES) algorithm (Worden et al., 2006, 2013b; Fu et al., 2013, 2016, 2018, 2019), optimal estimation approach (Rodgers, 2000) based on the Aura Tropospheric Emission Spectrometer (TES) retrieval algorithm (Bowman et al., 2006) with enhancements that enable the use of radiances from either one or multiple instruments. MUSES uses a multi-step retrieval process to characterize an atmospheric profile: temperature, water vapor, surface

properties, trace gases and cloud optical depth and height, thus accounting for the radiative impact of clouds. The optimal estimation method provides the vertical sensitivity (i.e., the averaging kernel matrix) and estimates of the uncertainties due to noise and radiative interferences from other geophysical parameters such as temperature and water vapor as described in Sect. 2. We use aircraft in situ observations from the HIAPER Pole-to-Pole (HIPPO) and Atmospheric

Tomography Mission (ATom) campaigns as well as the National Oceanic and Atmospheric Administration (NOAA) Global Monitoring Laboratory (GML) Global Greenhouse Gas Reference Network Aircraft Program (hereafter referred to simply as NOAA GML), taken between



2006 and 2017. The aircraft measurements, described in Sect. 2, span a wide range of latitudes, and include observations made over both ocean and land. Our validation methodology is described

in Sects. 3 and 4 and closely follows Oetjen et al. (2014) and Kulawik et al. (2021) and includes an evaluation of actual errors and a comparison to theoretical errors. The evaluation of results is presented in Sect. 4.

## 2. Data


## 2.1 Aircraft Data

Data from all five HIPPO aircraft missions (Wofsy et al., 2012) are used in this study: HIPPO-1 in January 2009; HIPPO-2 in October–November 2009; HIPPO-3 in March–April 2010; HIPPO-4 in June–July 2011; and HIPPO-5 in August–September 2011. During HIPPO, the National

Science Foundation's Gulfstream V flew tracks (Fig. 1) that were primarily over the Pacific Ocean but also crossed over New Zealand, Australia, and western North America at latitudes from 67S to 87N. The aircraft made steep ascents and descents along the flight path to construct vertical profiles approximately every 220 km or 20 minutes. The profiles had an average top of approximately 290 hPa. CO was measured with a quantum cascade laser spectrometer (QCLS) at

1 Hz frequency with accuracy of 3.5 ppb and $1\sigma$ precision of 0.15 ppb (McManus et al., 2010; Santoni et al., 2014). The QCLS CO measurements were compared with NOAA flask measurements over 59 HIPPO profiles and had a bias of -1.94 ppb which is within the accuracy estimate of the QCLS instrument (Santoni et al., 2014). HIPPO QCLS data have also been used to validate MOPITT satellite retrievals of CO (Deeter et al., 2013; Martínez-Alonso et al., 2014).

Data from ATom aircraft campaigns 1–3 (Wofsy et al., 2018) are also used in this study: ATom-1, July–August 2016; ATom-2, January–February 2017; and ATom-3 September–October 2017. During Atom, the NASA DC-8 aircraft flew tracks with similar latitude coverage as HIPPO but also flew over both the Atlantic and Pacific Oceans (Fig. 1). During flights, the aircraft continuously profiled the atmosphere from 0.2 to 12 km altitude with a similar average top to that

of HIPPO. For this study, we use CO measurements on ATom from the QCLS instrument, similar to HIPPO, that are calibrated to the WMO X2014A scale (Novelli et al., 1991, 1994, 1998).

The NOAA GML observations are taken mainly at fixed sites in North America (Sweeney et al., 2015). In this study observations from the years 2006–2017 and from nine sites (Fig. 1) are used. The air samples are collected using an automated Programmable Flask Package (PFP) operated on



small aircraft. Air samples are collected at several altitudes during a single flight resulting in a vertical profile for each trace gas measured. The average top of the profiles in the data set used here was at 440 hPa. The CO mixing ratios are reported relative to the WMO X2014A CO scale. Uncertainties on the CO from the flasks are of the order of 1 ppb (Sweeney et al., 2015).

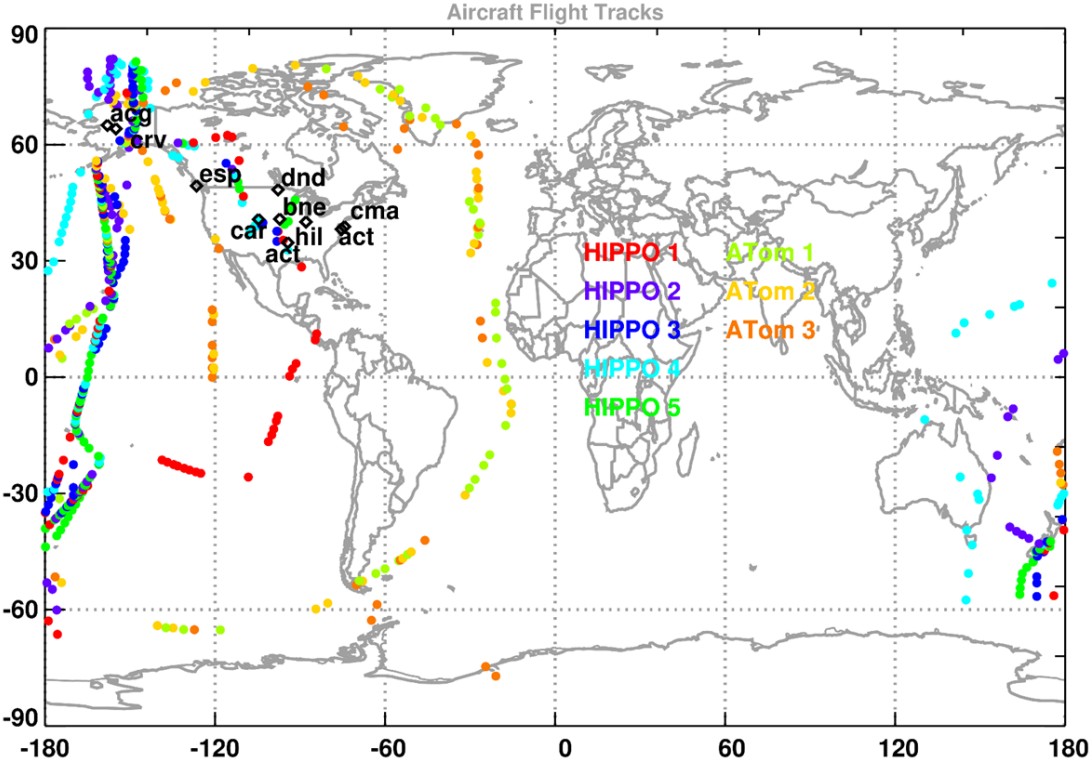


**Figure 1: Flight tracks for HIPPO, ATom (colored dots) and NOAA GML aircraft spiral locations as black diamonds with 3-character string identifier. Most NOAA GML site codes represent the site name (e.g., "cma" stands for offshore Cape May, New Jersey) while some site codes such as "act" and "crv" represent NOAA flask data collected on campaigns.**





### 2.3 AIRS single footprint CO retrievals

The Atmospheric Infrared Sounder (AIRS) is a nadir-viewing, scanning thermal infrared (TIR) spectrometer launched on board the Aqua satellite on May 4, 2002 into a sun synchronous polar orbit at an altitude of 705 km with 1:30 am and 1:30 pm local equator crossing times (Aumann et al., 2003). It measures the thermal radiance between 3–12 microns with a spectral resolution of ~ 1.8 cm$^{-1}$ in the 4.6 micron (~ 2100 cm$^{-1}$) CO absorption region. A single AIRS field of view (FOV) has a circular footprint with ~ 15 km diameter at nadir and the AIRS swath width is ~1650 km which enables near global coverage twice daily.

Several algorithm evaluations have been published previously for retrievals of CO from AIRS, using Level 2 cloud-cleared radiances (Susskind et al., 2003) on the 45 km fields of regard (FORs), which encompass nine FOVs. These include the AIRS operational algorithm (first introduced by McMillan et al. (2005) with revisions through to the current v7), the NOAA-Unique Combined Atmospheric Processing System (NUCAPS) (Gambacorta et al., 2015), the Community Long-term Infrared Microwave Combined Atmospheric Product System (CLIMCAPS) (Smith and Barnet, 2020) and the optimal estimation algorithm presented by Warner et al. (2010).

Here we present results of CO retrievals from AIRS radiances using the MUSES algorithm (Worden et al., 2006, 2013b; Fu et al., 2013, 2016, 2018, 2019; Kulawik et al., 2021). MUSES uses an optimal estimation approach (Rodgers, 2000) and leverages the algorithm developed for the Aura TES (Bowman et al., 2006). We use L1B radiances on single 15 km AIRS FOVs or footprints rather than cloud cleared radiances on the 45 km FORs (comprised of 9 FOVS) in order to preserve the original well-characterized radiance noise characteristics for use in our estimates (Irion et al., 2018; DeSouza-Machado et al., 2018). The Optimal Spectral Sampling (OSS) code was used as the forward model (Moncet et al., 2008, 2015). CO is retrieved using the 2181–2200 cm$^{-1}$ spectral range.

### 3. Validation Methodology

### 3.1 Coincidence criteria and quality control

The AIRS and aircraft profiles were matched using time and distance coincidence criteria of 9 hours and 50 km. The matched profiles were then subject to several quality control filters to form the final validation set: the aircraft profiles were required to have at least 10 pressure levels with valid CO data and the difference between the maximum and minimum pressure of the valid data



levels had to be at least 400 hPa. The AIRS MUSES algorithm provides diagnostic retrieval quality flags, and these were used to remove poor or suspect retrievals from the set. Since the AIRS MUSES algorithm uses the original non cloud-cleared radiances, profiles with thick clouds were also removed from the set. The cloud screening required that the average cloud effective optical depth over the AIRS spectrum and within the CO absorption band be less than 0.1. After

the quality screening was applied there remained 3734 AIRS – HIPPO matches representing 405 unique HIPPO aircraft profiles, 1324 AIRS – ATom matches representing 158 unique ATom aircraft profiles and 10044 AIRS - NOAA GML matches representing 747 unique NOAA GML aircraft profiles. Thus, each aircraft profile was evaluated against a set of AIRS profiles. All the aircraft profiles in the final data sets were interpolated vertically to the AIRS MUSES forward

model levels.

### 3.2 Approach for Error Validation

Details of the retrieval error characterization from the optimal estimation (OE) approach of Rogers

(2000) and its application to instruments like AIRS are provided in many publications (e.g., Boxe et al., 2010; Oetjen et al., 2014; Kulawik et al., 2021). Here the details relevant to the error validation in this study are presented.

As described in Oetjen et al. (2014) the OE error covariance can be split up into several terms, as shown in Equation (1), that represent the various factors contributing to the overall uncertainty $\mathbf{S}_z$

of a retrieved CO profile. These factors include smoothing due to limited vertical information content of the satellite instrument measurement (*smoothing*), instrument measurement noise (*noise*), uncertainties from parameters not included in the retrieval state vector (*systematic*), coupling interference or cross correlation between parameters retrieved simultaneously with CO (*cross-state*), and a residual term (*res*) that accounts for uncertainties not considered or unknown.


$$\mathbf{S_Z} = \underbrace{(\mathbf{A}_{ZZ} - \mathbf{I})\mathbf{S}_S(\mathbf{A}_{ZZ} - \mathbf{I})^T}_{smoothing} + \underbrace{\mathbf{GS}_e\mathbf{G}^T}_{noise} + \underbrace{\sum \mathbf{GK}_b\mathbf{S}_b(\mathbf{GK}_b)^T}_{systematic} + \underbrace{\sum \mathbf{A}_{xS}\mathbf{S}_a^{bret}(\mathbf{A}_{xS})^T}_{cross\text{-}state} + res \quad (1)$$

In the *smoothing* term $\mathbf{I}$ is the identity matrix, $\mathbf{A}_{zz}$ is the covariance matrix for CO and $\mathbf{S}_s$ is the

smoothing error covariance. In the *noise* term $\mathbf{G}$ is the gain matrix that describes the sensitivity of the retrieved state to changes in measured radiances and $\mathbf{S}_e$ is the instrument noise covariance. In the *systematic* term the subscript $b$ represents parameters that are held constant during the retrieval





with respective Jacobians, $\mathbf{K}_b$, and error covariance matrix $\mathbf{S}_b$. In the *cross-state* term the averaging
kernels of the other parameters ($x$) retrieved simultaneously with CO are $\mathbf{A}_{xs}$ and the corresponding
error covariance matrix is $\mathbf{S}_a^{bret}$.

The averaging kernel matrix describes the vertical sensitivity of a retrieved parameter to its true
state in the atmosphere. The vertical sensitivity is dependent on the true state vertical distribution
of CO and other trace gases, retrieval constraints, and on the interference of other geophysical
parameters such as the profiles of temperature and water vapor. The sum of the rows of the
averaging kernel matrix provides information on the location of the peak sensitivity of the retrieval.
Figure 2 shows the mean sum of the rows of the averaging kernel matrices for all the AIRS profiles
in the validation set binned by latitude band: the level of peak sensitivity is generally between 400
and 500 hPa. The sensitivity peaks at a higher level in the tropical and sub-tropical latitude band
of 30S–30N and at lower vertical levels in the higher latitude bands of both hemispheres.

For comparing satellite profiles of trace gases with limited vertical resolution to profiles measured
in situ from aircraft, the averaging kernel and an a priori profile is applied to the in situ profiles as
in Rodgers and Connor (2003). Through this procedure a new profile $\hat{\mathbf{Z}}$, representing what the
satellite "sees" assuming no retrieval errors, is generated as shown in Equation (2) from the
averaging kernel $\mathbf{A}_{zz}$ applied to the difference between the elements of the original aircraft profile
$\mathbf{Z}_{aircraft}$ and the a priori profile $\mathbf{Z}_{apriori}$.

$$\hat{\mathbf{Z}} = \mathbf{Z}_{apriori} + \mathbf{A}_{zz} ( \mathbf{Z}_{aircraft} - \mathbf{Z}_{apriori} ) \qquad\qquad (2)$$

This procedure is also referred to as convolving the in situ profiles with the averaging kernel.
Since there are no aircraft observations for the part of the retrieved profile above the aircraft flight
levels, numerical techniques must be applied to extrapolate aircraft profiles above the flight levels
(e.g. Kulawik et al., 2021); however, the uncertainty of the extrapolated measurements at these
levels must be accounted for as it can propagate to the levels where there is actual aircraft
observations through the application of the averaging kernel (Tang et al., 2020). For our study we
simply fill the true aircraft profile above the aircraft flight levels with the a priori value. If the a
priori is representative of the average true atmosphere this assumption should be reasonable. We
explore the implications of this assumption using the NOAA GML set in Sect. 4.3.

The approach for error validation in this paper will start with a comparison of each AIRS retrieved
profile with the corresponding matched aircraft profile convolved with the averaging kernel; the

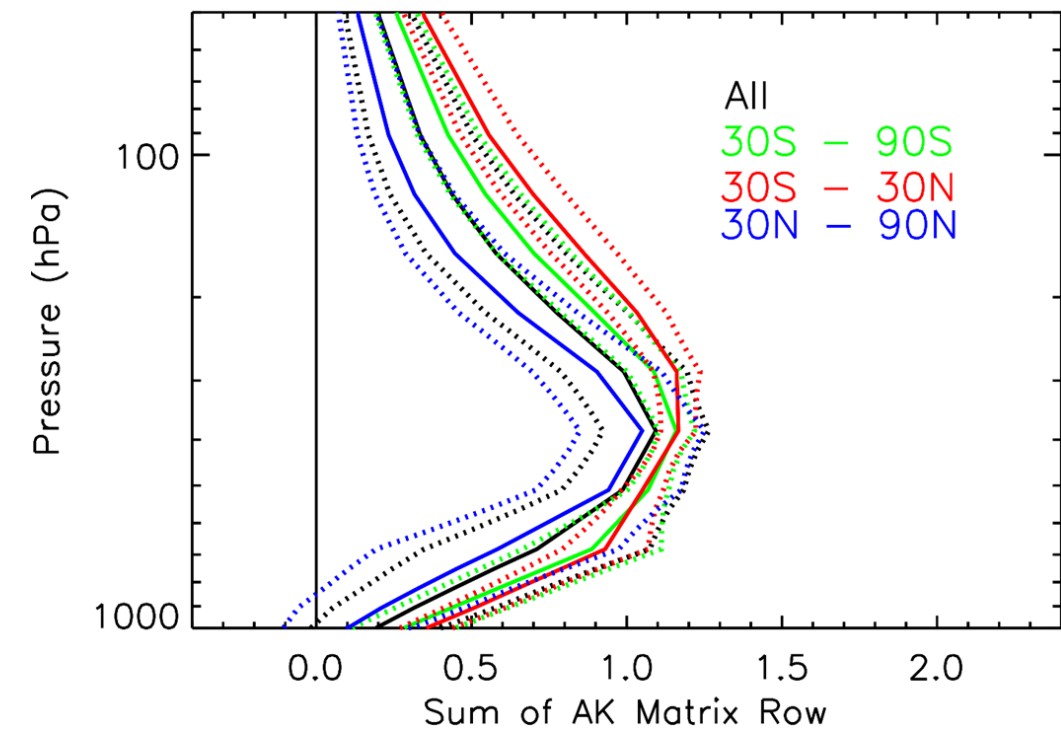


**Figure 2: Mean (solid) sum of rows of the AIRS MUSES CO averaging kernels for each latitude band for the HIPPO retrievals. The dotted lines are one standard deviation from the mean. The peak of the mean generally corresponds to the vertical level of maximum AIRS sensitivity to the true state CO mixing ratio.**





results will be grouped in latitude bands ranging from the tropics to sub-polar regions. Next theoretical errors represented by all but the *smoothing* term of the error covariance of Eq. (1) will be evaluated for each retrieval, averaged within the latitude bands, and compared to the retrieval

error standard deviation (uncertainty) and the a priori error. Finally, empirical errors calculated from an ensemble of retrieved profiles collocated with an aircraft profile as in Boxe et al. (2010) and Oetjen et al. (2014) will be evaluated for select CO plume and background cases. This approach will be applied separately to the HIPPO, ATom and NOAA datasets, since each presented different characteristics.


## 4. Results

### 4.1 AIRS MUSES validation with HIPPO

The percent differences between AIRS MUSES and the HIPPO aircraft profiles are shown in Fig. 3. The profiles are plotted only up to 200 hPa, as there were few aircraft observations above that level, and are shown as the complete set and binned by latitude bands. For all groupings the mean biases are positive in the lower troposphere, tend toward zero in the middle troposphere, where the retrieval has greatest sensitivity, and become negative in the upper troposphere. The spread of the

error profiles also tends to be narrower in the middle of the troposphere. Table 1 shows statistics corresponding to these plots and for the profiles grouped by land/ocean and day/night categories for selected pressure levels. The lowest biases are within plus or minus 3.1% and occur at the 510 hPa level while there are larger positive biases between 2–21% at the 750 hPa level and negative biases up to ~ 15% at the 287 hPa level. There were no substantial or consistent differences for

the error statics grouped by land vs ocean and day vs night, which suggests that these categories can be combined in the error analysis. Partial column average mixing ratios (referred to hereafter as column average mixing ratios) were calculated for each profile between the lowest to the highest aircraft flight level. The column average CO mixing ratios plotted by latitude (Fig. 4, top panel) show that the 30S–90S band was predominantly in a background regime with mixing ratios

generally < 70 ppbv, and that mixing ratios increased steadily with latitude to ~ 150 ppbv by 30N. The average column CO mixing ratio bias (Fig. 4 bottom panel) also shows a latitude dependence with higher mean bias of ~ 10–15 ppbv occurring near 30N band. In addition, the error distribution is highly skewed toward positive numbers particularly in the 30N–60N latitude band (skewness=1.36), indicating that the errors are not normally distributed.

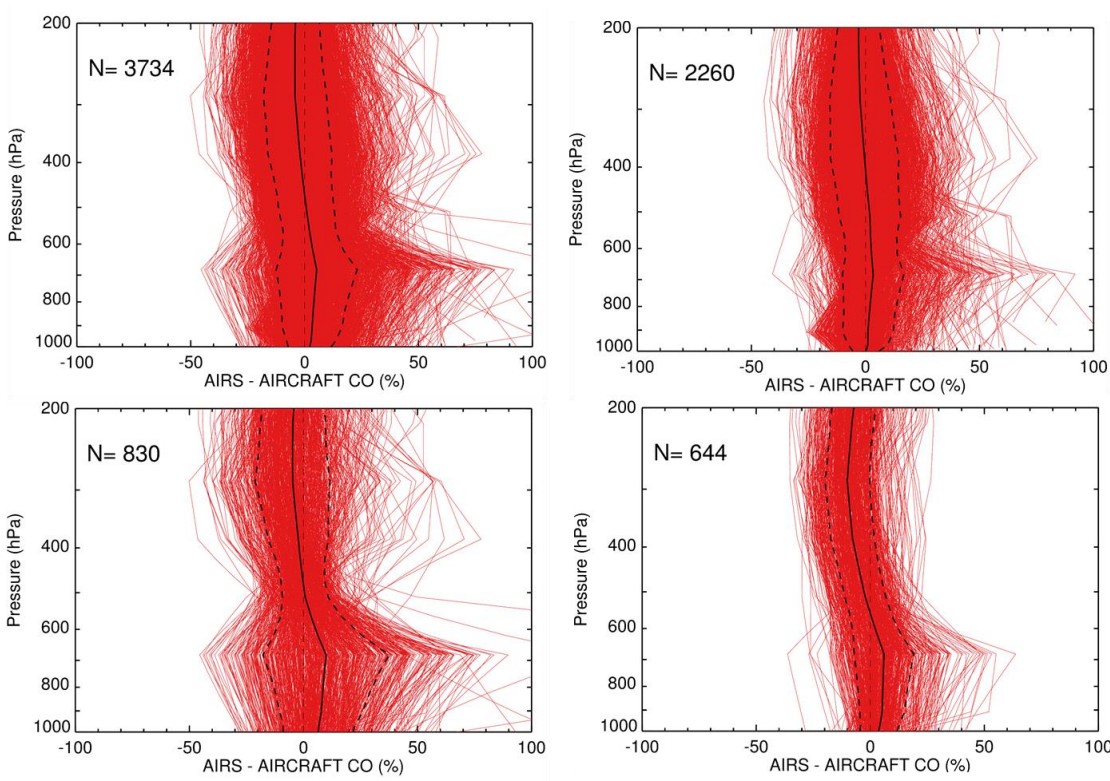


**Figure 3: The AIRS MUSES-Aircraft percent difference profiles for HIPPO. The number of profiles and the latitude bands are indicated in the upper left. All HIPPO profiles were convolved with the averaging kernels (Eq. 2) before the differences were calculated. The red lines indicate the individual profiles, the black solid lines the mean difference or bias, and the dashed lines one standard deviation from the mean.**





**Table 1:  AIRS – Aircraft statistics for HIPPO campaign.**

|  | Bias 749.89 hPa (%) | STD 749.89 hPa (%) | Bias 510.90 hPa (%) | STD 510.89 hPa (%) | Bias 287.30 hPa (%) | STD 287.30 hPa (%) | Bias Column (%) | STD Column (%) | N Profs |
|---|---|---|---|---|---|---|---|---|---|
| **All** | 4.56 | 15.69 | 0.95 | 12.19 | -4.22 | 13.86 | 0.69 | 9.20 | 3734 |
| **30 S – 30 N** | 9.09 | 23.71 | 0.77 | 10.00 | -4.75 | 16.40 | 2.37 | 10.56 | 830 |
| **30 N – 90 N** | 2.56 | 12.22 | 1.94 | 13.48 | -2.35 | 13.35 | 0.62 | 8.99 | 2260 |
| **30 S – 90 S** | 5.72 | 11.76 | -2.29 | 8.99 | -10.10 | 9.73 | -1.21 | 7.50 | 644 |
| **Land** | 6.96 | 15.67 | -0.12 | 12.22 | -7.52 | 12.96 | -0.32 | 8.96 | 930 |
| **30 S – 30 N** | 21.07 | 24.69 | 0.38 | 8.21 | -14.58 | 14.84 | 3.34 | 8.60 | 37 |
| **30 N – 90 N** | 5.30 | 13.90 | -0.88 | 11.80 | -6.96 | 12.12 | -0.42 | 9.05 | 799 |
| **30 S – 90 S** | 3.90 | 10.53 | -0.59 | 11.31 | -7.55 | 11.54 | -0.92 | 8.03 | 94 |
| **Ocean** | 3.76 | 15.62 | 1.31 | 12.16 | -3.13 | 13.98 | 1.03 | 9.26 | 2804 |
| **30 S – 30 N** | 8.53 | 23.53 | 0.78 | 10.08 | -4.29 | 16.34 | 2.32 | 10.64 | 793 |
| **30 N – 90 N** | 0.32 | 9.39 | 3.05 | 13.87 | 0.29 | 12.82 | 1.19 | 8.92 | 1461 |
| **30 S – 90 S** | 6.03 | 11.95 | -2.58 | 8.51 | -10.53 | 9.33 | -1.26 | 7.41 | 550 |
| **Day** | 4.70 | 15.18 | 0.10 | 12.01 | -5.13 | 13.98 | -0.11 | 8.73 | 1785 |
| **30 S – 30 N** | 9.32 | 23.08 | -1.13 | 10.86 | -7.84 | 17.54 | 1.03 | 10.18 | 256 |
| **30 N – 90 N** | 3.62 | 13.83 | 0.86 | 12.76 | -3.47 | 13.87 | -0.02 | 8.63 | 1210 |
| **30 S – 90 S** | 5.08 | 10.80 | -1.80 | 9.43 | -9.28 | 9.24 | -1.37 | 7.68 | 319 |
| **Night** | 5.30 | 16.88 | 1.16 | 11.93 | -4.29 | 13.67 | 1.39 | 9.80 | 1723 |
| **30 S – 30 N** | 8.99 | 24.01 | 1.61 | 9.48 | -3.38 | 15.69 | 2.96 | 10.68 | 574 |
| **30 N – 90 N** | 2.32 | 10.64 | 2.40 | 14.11 | -2.33 | 12.55 | 1.26 | 9.82 | 824 |
| **30 S – 90 S** | 6.34 | 12.63 | -2.78 | 8.52 | -10.90 | 10.14 | -1.06 | 7.32 | 325 |


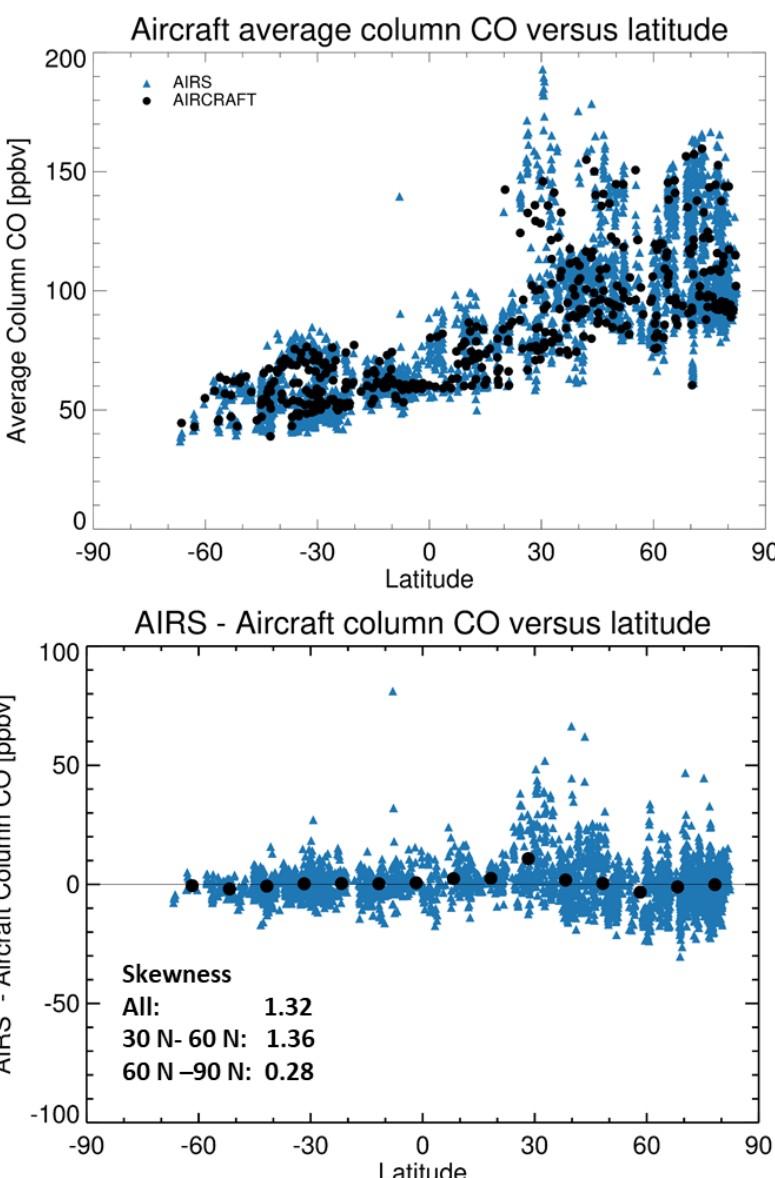

**Figure 4: The AIRS and HIPPO partial column average mixing ratios (top) and AIRS – HIPPO column average mixing ratio differences (bottom) by latitude. The column averages are calculated from the lowest to the highest flight altitudes for each profile. The black dots in the bottom figure are the average differences within each 10-degree latitude bin. The skewness of the error distribution is also shown. Skew values greater (less) than 1 indicate significant positive (negative) statistics skew from a Gaussian distribution.**



Beyond examining biases and variability of the retrieved profiles, evaluating the retrieval error
        estimates is also important, since they provide users with a measure of the reliability of the data.
        Following Oetjen et al. (2014) and Kulawik et al. (2021) we evaluated the AIRS MUSES retrievals
        by comparing the theoretical error estimates from the MUSES diagnostics to the retrieval error
        statistics described above.  Figure 5 shows the profiles of the fractional estimated observation

errors, mean a priori error, AIRS-aircraft standard deviation, and a priori–aircraft standard
        deviation.  The errors are binned by latitude band and the 30–90-degree bands have been divided
        into two bands of 30–60 and 60–90 degrees in both hemispheres to better capture the dependence
        of error characteristics on latitude.   The estimated observational error includes the noise,
        systematic and cross-state error terms as shown in Eq. (1) and the mean a priori error is estimated

from the square root of the diagonal of the a priori covariance matrix.
        The estimated theoretical observational errors (red lines are individual errors and the blue lines are
        the mean) are lowest around 500 hPa where AIRS sensitivity is greatest, and this pattern is similar
        to the actual error profiles shown in Fig. 3. The minimum error shifts downwards towards the poles
        with the smallest errors occurring lower at about 650 hPa in the Arctic region 60N–90N; however,

in the Antarctic region (60S–90S) there were not enough AIRS-aircraft profile matches where the
        AIRS profiles passed quality screening to provide a reasonable set of statistics.
        The standard deviation for the a priori-aircraft differences (green) is lower than the standard
        deviation for the AIRS-aircraft differences (black); for this dataset the a priori appears to be a
        better estimate of the truth than the retrieval; however, the skewness of the column mixing ratio

differences suggests that Gaussian statistics do not provide an accurate representation of the error
        characteristics of this dataset, i.e., a simple average of error estimates is not very meaningful. Note
        also the average estimated error (blue) is significantly lower than the AIRS-aircraft differences
        (black) except below 600 hPa in the 30S–60 N range, which is also likely due to the skewness of
        the data differences.

An alternative approach for evaluating the theoretical error is to compare it to the variability within
        the set of AIRS profiles collocated with an aircraft profile, which can be thought of as an empirical
        error (Oetjen et al., 2014). Using this approach, plume and background cases were selected for
        each of the five HIPPO missions.  The case profiles were chosen using the maximum and minimum
        CO mixing ratios for each campaign at the 464.16 hPa pressure level of the remapped aircraft




**Figure 5:** **Estimated observational error analysis for the HIPPO data set. Estimated observation errors for each AIRS MUSES CO retrieval (dotted red lines), the mean observation error (solid blue line and triangles) the mean a priori error estimate (green line) and the standard deviation of the AIRS MUSES – HIPPO aircraft profiles differences and the standard deviation of the a priori – aircraft profile differences. The profiles are binned by latitudes bands 30N–60N, 60N–90N, 30S–30N, 30S–60S and 60S–90S.**



profiles. In addition to the CO mixing ratio criteria a minimum of eight co-located AIRS profiles that met the quality control standards had to be available for the case to be selected. A mean theoretical error for this set of co-located AIRS profiles is estimated using the same terms in Equation 1 as those for the observation errors of Fig. 5. The empirical error was estimated as the square root of the diagonal of the covariance matrix of all the coincident AIRS MUSES retrievals. In general, for these cases, the empirical errors were of the same magnitude as the theoretical errors

and the absolute differences were less than 10%. For the background cases the empirical errors are generally comparable to the theoretical errors. For the plume cases, we might expect to see larger discrepancies between theoretical and empirical errors, due to atmospheric variability in the region of the plume.

Illustrative cases for HIPPO-2 and HIPPO-3 are presented in Fig. 6. The plume case for HIPPO-

2 is in the Arctic; the aircraft data feature a very high spike (~270 ppb) near 400 hPa that the mean AIRS profile does not capture (Fig. 6 bottom left panel). The empirical error has a large peak > 15% at about the same level that is much larger than the theoretical value (Fig. 6 top left panel). For the HIPPO-3 plume case the observed CO is also high with peaks greater than 200 ppb in the middle troposphere. In this case, the AIRS mean retrieval does capture a peak (Fig. 6 right bottom

panel), and the empirical and theoretical errors are in reasonable agreement.

### 4.2 AIRS MUSES validation with ATom

The same steps were followed for the analysis of the ATom dataset. The percent differences

between AIRS MUSES and the ATom aircraft profiles are shown in Fig. 7 for different latitude bands and the error statistics corresponding to these plots are shown in Table 2. As with HIPPO the smallest biases are in the middle troposphere and cover a similar range (from ~ -4 to + 5 % vs -3 to +3%). The average column mixing ratios (Fig. 8) show the same dependence on latitude, as do the column errors. Like the HIPPO set, the estimated observational errors (Fig. 9) were smallest

in the middle troposphere. However, the standard deviation of the AIRS – aircraft differences is smaller for the ATom comparisons than for the HIPPO comparisons. In the vertical range where AIRS has good sensitivity to CO (~600 to 200 hPa), The standard deviation of the AIRS – ATom differences is generally less than the standard deviation of the a priori – ATom differences, except south of 30S, where there are mostly low levels of CO. The distribution of errors in the 30N–60N

latitude band is less skewed than for HIPPO (0.54 vs. 1.36) suggesting that



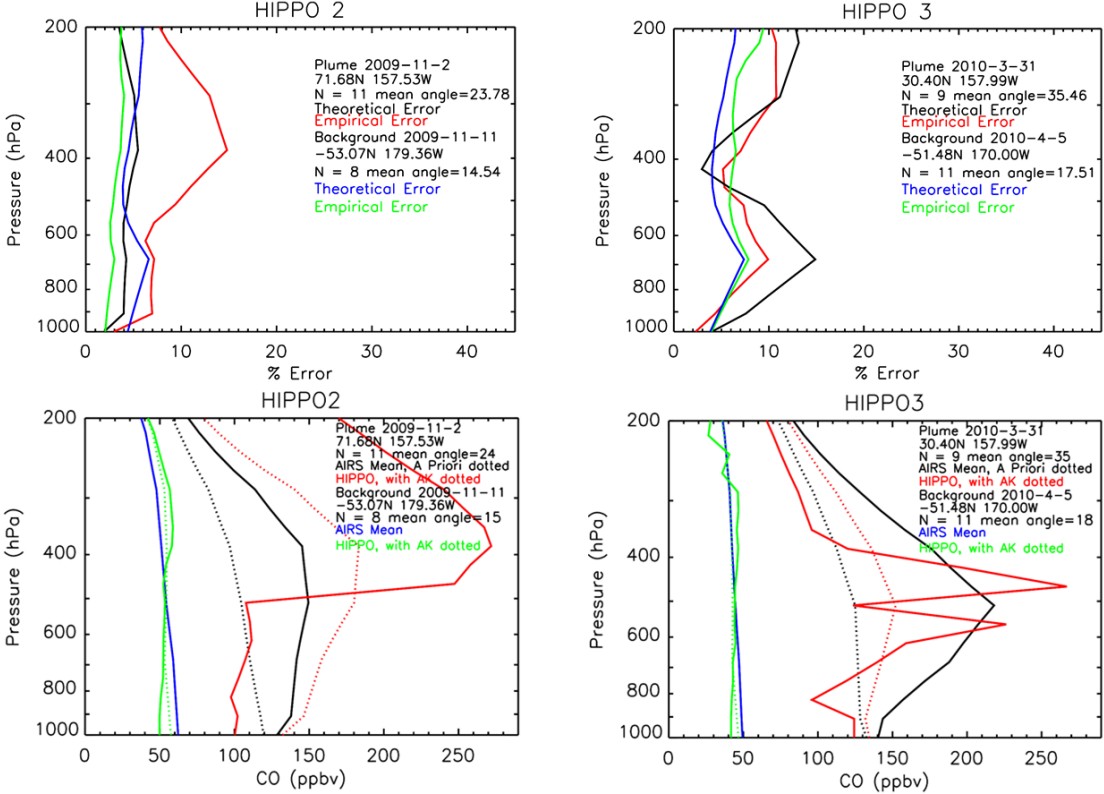

**Figure 6: Theoretical and empirical errors for selected plume and background cases from the HIPPO campaign (top panels). Theoretical errors are black (plume profiles) and blue (background profiles) and empirical errors are red (plume profiles) and green (background profiles). Theoretical errors include do not include the smoothing term. In the bottom panels the plume (red) and background (green) HIPPO and average AIRS profiles (plume black, background blue) corresponding to the theoretical and empirical error profiles in top panels are shown. The HIPPO profiles are shown without (solid) and with (dotted) the application of the AIRS averaging kernel. The average AIRS a priori profiles are shown for the plume cases only as black dots.**



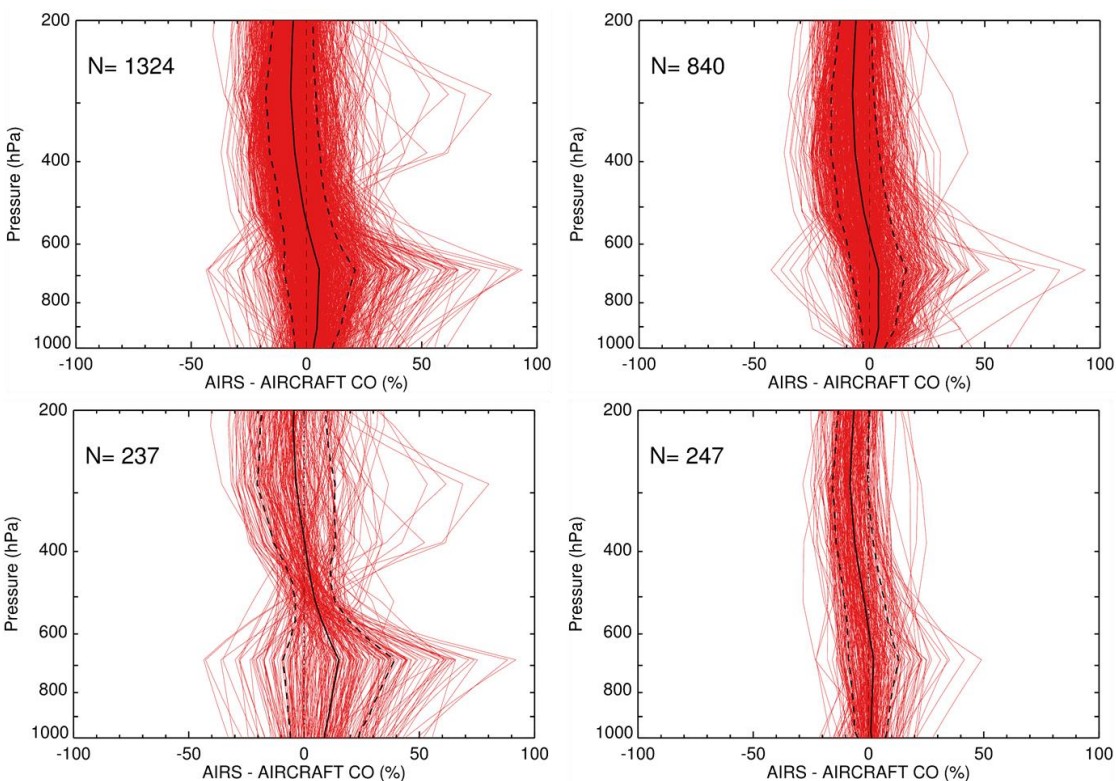

**Figure 7: The AIRS MUSES- Aircraft percent difference profiles for ATom campaigns 1-3. The number of profiles and the latitude bands are indicated in the upper left. All ATom profiles were convolved with the averaging kernels (Eq. 2) before the differences were calculated. The red lines indicate the individual profiles, the black solid lines the mean difference or bias, and the dashed lines one standard deviations from the mean.**






Table 2: AIRS – Aircraft statistics for ATom campaigns 1-3.

| | Bias 749.89 hPa (%) | STD 749.89 hPa (%) | Bias 510.90 hPa (%) | STD 510.89 hPa (%) | Bias 287.30 hPa (%) | STD 287.30 hPa (%) | Bias Column (%) | STD Column (%) | N Profs |
|---|---|---|---|---|---|---|---|---|---|
| **All** | 5.19 | 13.40 | -1.10 | 10.58 | -6.84 | 10.91 | 0.02 | 8.26 | 1324 |
| **Pacific** | 4.46 | 11.80 | -2.90 | 10.35 | -7.55 | 9.32 | -1.14 | 7.33 | 708 |
| **30 S – 30 N** | 13.46 | 21.70 | 4.81 | 8.62 | -3.49 | 16.93 | 6.22 | 8.34 | 237 |
| **30 N – 90 N** | 3.83 | 10.01 | -2.55 | 11.00 | -7.43 | 9.25 | -1.07 | 7.69 | 840 |
| **30 S – 90 S** | 1.84 | 9.44 | -1.81 | 8.77 | -8.03 | 7.69 | -2.21 | 7.21 | 247 |
| **Land** | 1.98 | 8.69 | -2.78 | 9.55 | -6.41 | 9.58 | -1.94 | 7.16 | 349 |
| **30 S – 30 N** | NA | NA | NA | NA | NA | NA | NA | NA | 0 |
| **30 N – 90 N** | 1.94 | 8.65 | -2.82 | 9.65 | -6.40 | 9.80 | -1.98 | 7.13 | 326 |
| **30 S – 90 S** | 2.46 | 9.52 | -2.06 | 8.18 | -6.58 | 5.54 | -1.44 | 7.64 | 23 |
| **Ocean** | 6.33 | 14.56 | -0.50 | 10.87 | -6.99 | 11.34 | 0.72 | 8.51 | 975 |
| **30 S – 30 N** | 13.46 | 21.70 | 4.81 | 8.62 | -3.49 | 16.93 | 6.22 | 8.34 | 237 |
| **30 N – 90 N** | 5.02 | 10.63 | -2.38 | 11.78 | -8.09 | 8.83 | -0.50 | 7.98 | 514 |
| **30 S – 90 S** | 1.77 | 9.45 | -1.80 | 8.84 | -8.18 | 7.87 | -2.29 | 7.17 | 224 |
| **Day** | 4.56 | 12.17 | -0.57 | 10.35 | -5.80 | 10.86 | 0.10 | 7.49 | 734 |
| **30 S – 30 N** | 10.17 | 21.07 | 5.65 | 10.10 | -0.07 | 19.17 | 5.39 | 8.20 | 99 |
| **30 N – 90 N** | 4.15 | 9.58 | -1.68 | 10.13 | -6.77 | 8.68 | -0.52 | 6.62 | 512 |
| **30 S – 90 S** | 1.78 | 10.72 | -0.95 | 9.74 | -6.39 | 8.17 | -1.56 | 8.48 | 123 |
| **Night** | 6.36 | 15.21 | -1.39 | 10.87 | -8.12 | 10.94 | 0.24 | 9.19 | 546 |
| **30 S – 30 N** | 15.82 | 21.90 | 4.20 | 7.36 | -5.95 | 14.71 | 6.81 | 8.42 | 138 |
| **30 N – 90 N** | 3.72 | 11.23 | -3.54 | 12.47 | -8.51 | 10.27 | -1.60 | 9.27 | 284 |
| **30 S – 90 S** | 1.89 | 8.01 | -2.68 | 7.62 | -9.66 | 6.83 | -2.85 | 5.63 | 124 |




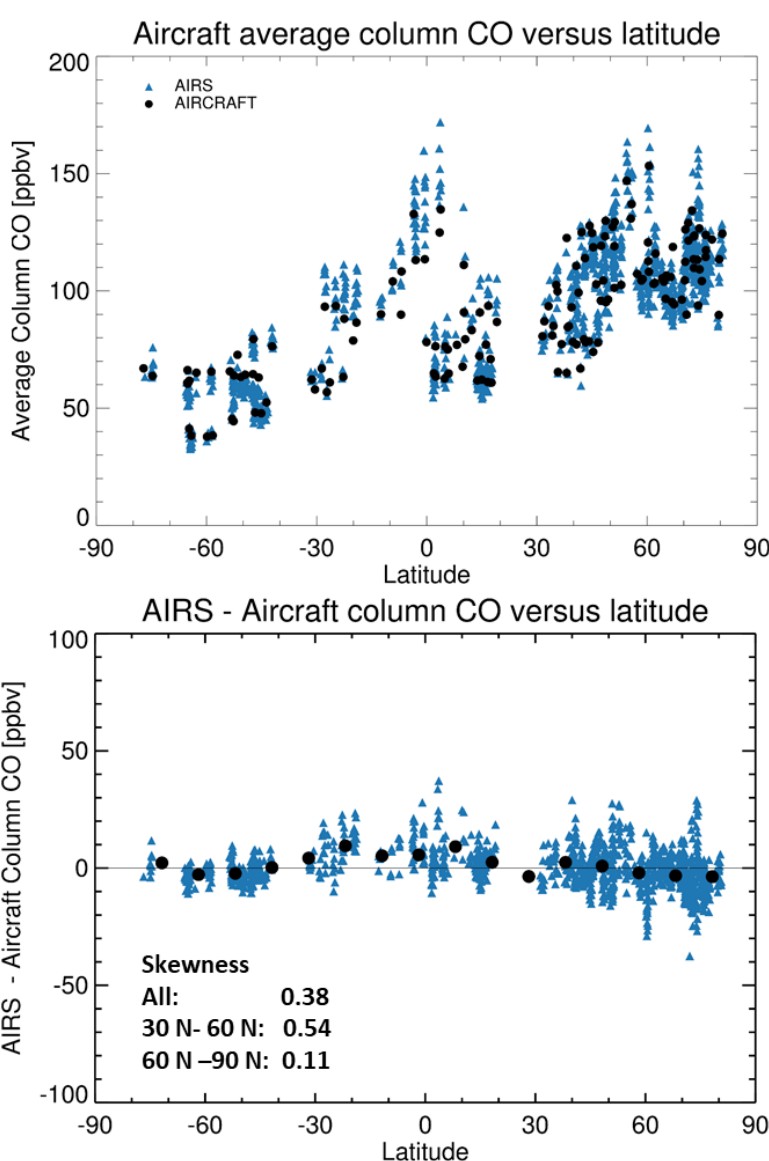

**Figure 8: The AIRS and partial column average mixing ratios (top) and AIRS – ATom column average mixing ratio differences (bottom) by latitude. The column averages are calculated from the lowest to the highest flight altitudes for each profile. The black dots in the bottom figure are the average differences within each 10-degree latitude bin.**



a Gaussian distribution of errors is a reasonable assumption for this dataset. The difference between HIPPO and ATom was most evident in the 30N–60 N band where for HIPPO the retrieval
error standard deviation was ~ 4 % larger than the a priori error standard deviation (Fig. 6) whereas for ATom the retrieval error standard deviation was ~ 5 % smaller than the a priori error standard deviation.

The reason for the "better" retrieval performance relative to the prior for the ATom vs the HIPPO comparisons is not immediately clear. In the 30N–60N latitude band, the mean and standard
deviation of the average column CO amounts for HIPPO and ATom were similar at 103 and 108 ppb and 409 and 445 ppb respectively. The datasets have similar seasonal coverage. There was a significant difference in geographic coverage: the HIPPO flights only covered the Pacific Ocean and adjacent land whereas ATom additionally flew over the Atlantic Ocean (Fig. 1). To determine if this difference influenced the statistics a subset of the ATom data set was generated that
considers only points west of 75W longitude. The statistics for this case are shown in Table 2 in the row labeled "Pacific". While the bias at 510 hPa is slightly more negative for the Pacific case at -2.98% compared to -1.10% for all cases, the standard deviation of the AIRS-aircraft differences are similar. Furthermore, for the Pacific case there was no significant skew in the column average mixing ratio error distribution (30N–60N skewness=0.29) and the estimated observation error
profiles (not shown) were similar to those in Fig. 9. Therefore, it does not appear that the different geographic coverage between HIPPO and ATom was the cause of the differences in the error statistics.

Fig. 10 shows example comparisons of theoretical and empirical error estimates for selected ATom match-ups (as presented for HIPPO in Fig. 6). The plume in the ATom-1 example is retrieved at
a much higher altitude than observed and the empirical error is much higher than the theoretical error (Fig. 10 left panels), while in the ATom-2 example there is a better match between the retrieved and observed profiles, and the empirical and theoretical errors are comparable Overall, this analysis shows similar features to the one using the theoretical observation errors by latitude band in Fig. 9.


## 4.3 AIRS MUSES validation with NOAA GML

The NOAA GML dataset was much larger, spanning a much longer period (2006–2017), but provided results over only a limited number of locations in North America (Fig. 1). For the NOAA


**Figure 9:** **Estimated observational error analysis for the ATom data set. Estimated observation errors for each AIRS MUSES CO retrieval (dotted red lines), the mean observation error (solid blue line and triangles) the mean a priori error estimate (green line) and the standard deviation of the AIRS MUSES – ATom aircraft profiles differences and the standard deviation of the a priori – aircraft profile differences. The profiles are binned by latitudes bands 30N–6 N, 6 N–90N, 30S–30N, 30S–60S and 60S–90S.**



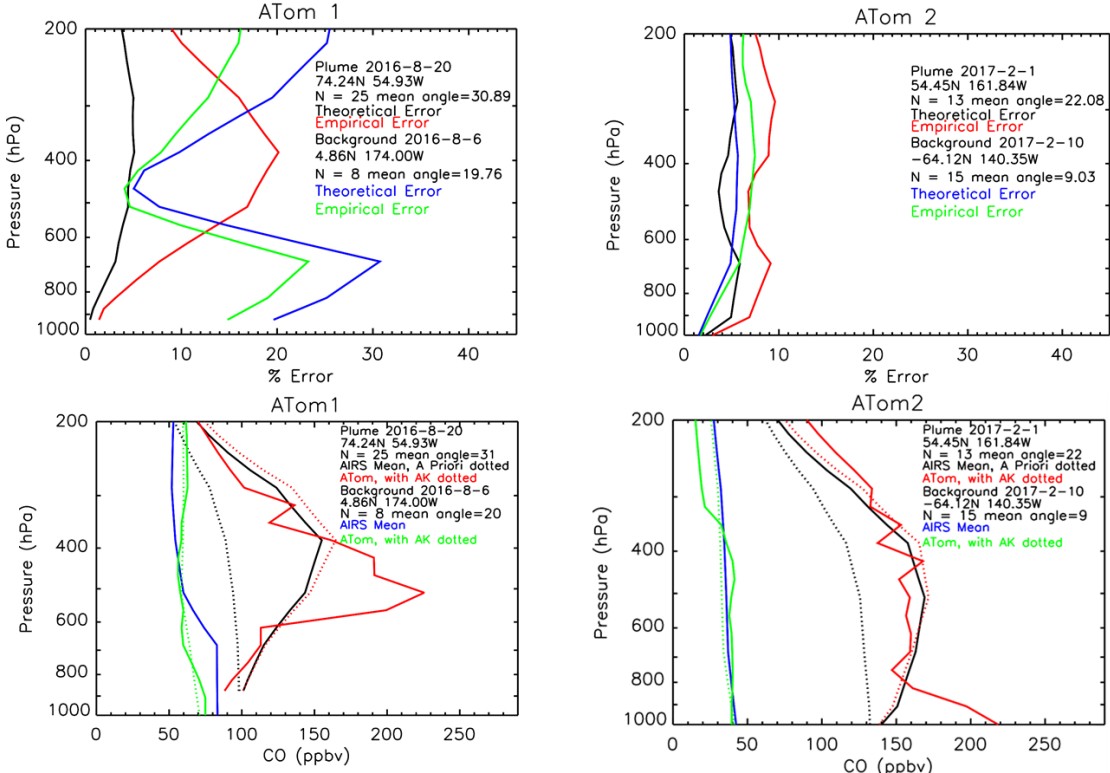

**Figure 10: Theoretical and empirical errors for selected plume and background cases from the ATom campaigns (top panels). Theoretical errors are black (plume profiles) and blue (background profiles) and empirical errors are red (plume profiles) and green (background profiles). Theoretical errors include do not include the smoothing term. In the bottom panels the plume (red) and background (green) ATom and average AIRS profiles (plume black, background blue) corresponding to the theoretical and empirical error profiles in top panels are shown. The ATom profiles are shown without (solid) and with (dotted) the application of the AIRS averaging kernel. The average AIRS a priori profiles for the plume cases only are shown as black dots.**



GML set the AIRS MUSES retrieval error profiles are shown in Fig. 11 and statistics are shown in Table 3. Table 3 indicates that there are about a third of the matched profiles listed as ocean points which seems to contradict the map in Fig. 1 that shows all the NOAA GML location over land. However, the land/ocean classification is based on the MUSES land/ocean flag and several

of the NOAA GML locations are at the coast and one, "cma", is identified as offshore Cape May. Therefore, a substantial number of the AIRS FOVs within the 50 km radius of the NOAA GML profiles near the coast and those corresponding to "cma" were classed as ocean. The column average mixing ratio errors by latitude are shown in Fig. 12. Overall, the retrievals have a noticeably larger positive bias in the lower troposphere compared to the HIPPO and ATom sets.

At the 510 hPa level the biases over land/ocean and day/night categories range from 4.9–9.6% for the NOAA GML set (Table 3) compared to less than plus or minus 4% for the HIPPO and ATom sets in the corresponding 30N–90 N latitude band (Tables 1 and 2). The column average mixing ratios are also biased much higher ranging from 7.2–10.7% for NOAA GML (Table 3) compared to within plus or minus 2% for the HIPPO and ATom sets (Tables 1 and 2). The higher biases

seem consistent across the latitudinal range of the NOAA GML observations as shown in Fig. 12. The theoretical observations errors for the NOAA GML set (Fig. 13) are similar to those of the HIPPO set (Fig. 5) with larger AIRS MUSES-aircraft error standard deviations than the mean observation errors and the a priori error standard deviations. As with HIPPO the column average mixing ratio errors are highly skewed toward positive values with an overall skewness of 1.57.

This suggests that the assumption of a Gaussian error distribution upon which the theoretical observational error analysis is based is also not valid for the NOAA GML set.

We hypothesized that the higher retrieval biases for the NOAA GML set may be an artifact of larger errors associated with extrapolation of the aircraft profiles above the uppermost measurement altitude. The NOAA GML profiles have an average highest flight level near 440 hPa

compared to 290 hPa for the HIPPO and ATom sets and therefore there are more retrieval levels to fill in the remapped aircraft profile. These extra fill levels can cause greater error uncertainty in the lower levels when the averaging kernel matrix is applied. Tang et al. (2020) found that errors in MOPITT aircraft CO comparisons were very sensitive in the middle and upper troposphere to the method used to extend the aircraft profile.

To test the sensitivity of the AIRS retrieval statistics to the mixing ratio values used to fill the aircraft profiles, an additional set of statistics was generated using a scaled a priori value to fill the



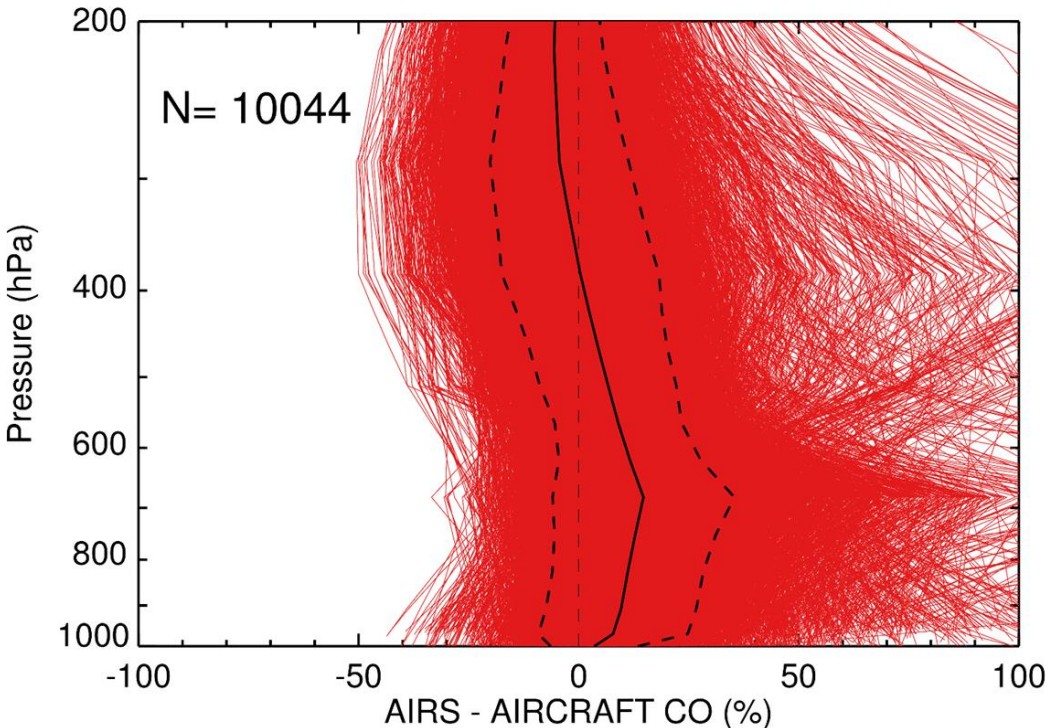

**Figure 11**: **The AIRS MUSES-Aircraft percent difference profiles for NOAA GML aircraft CO observations. All aircraft profiles were convolved with the averaging kernels (Eq. 2) before the differences were calculated. The red lines indicate the individual profiles, the black solid lines the mean difference or bias, and the dashed lines one standard deviations from the mean.**






**Table 3: AIRS – Aircraft statistics for the NOAA GML observations. By default, the aircraft are filled above the flight levels with the a priori profile. Additional statistics are generated by filling above the flight level with the a priori scaled by the difference between the a priori and the aircraft value at the top flight level (All Scale Fill).**

| | Bias 749.89 hPa (%) | STD 749.89 hPa (%) | Bias 510.90 hPa (%) | STD 510.89 hPa (%) | Bias 287.30 hPa (%) | STD 287.30 hPa (%) | Bias Column (%) | STD Column (%) | N Profs |
|---|---|---|---|---|---|---|---|---|---|
| **All** | 12.85 | 18.33 | 6.68 | 15.49 | -4.37 | 15.76 | 9.42 | 13.50 | 10044 |
| **Land** | 13.79 | 20.34 | 6.99 | 16.78 | -4.40 | 16.83 | 10.25 | 15.07 | 6534 |
| **Ocean** | 11.11 | 13.67 | 6.08 | 12.74 | -4.30 | 13.53 | 7.90 | 9.76 | 3510 |
| **Day** | 15.41 | 20.63 | 4.91 | 14.24 | -6.70 | 14.37 | 10.74 | 14.96 | 6289 |
| **Night** | 8.57 | 12.51 | 9.64 | 16.99 | -0.46 | 17.14 | 7.20 | 10.24 | 3755 |
| **All Scale Fill** | 9.82 | 18.22 | 0.67 | 15.09 | -10.21 | 15.45 | 5.75 | 13.31 | 10044 |












**Figure 12:** **The AIRS and NOAA ESRL partial column average mixing ratios (top) and AIRS – aircraft column average mixing ratio differences (bottom) by latitude. The column averages are calculated from the lowest to the highest flight altitudes for each profile. The black dots in the bottom figure are the average differences within each 10-degree latitude bin.**

**Figure 13: Estimated observational error analysis for the NOAA GML data set. Estimated observation errors for each AIRS MUSES CO retrieval (dotted red lines), the mean observation error (solid blue line and triangles) the mean a priori error estimate (green line) and the standard deviation of the AIRS MUSES – NOAA ESRL aircraft profiles differences and the standard deviation of the a priori – aircraft profile differences. The errors are based on the NOAA ESRL observations. The profiles are binned by latitudes bands 30N–60N, 60N–90N.**

aircraft profiles above the flight levels. The scaled a priori value used a constant scale ratio
between the mixing ratio at the highest aircraft level and the a priori at that level. The retrieval
statistics for this experiment are shown in the last row of Table 3. For the scaled a priori fill case
the bias at 510 hPa is only 0.7% but the column average mixing ratio bias is still large at 5.8%.
Clearly the choice of fill value has a large impact on the retrieval error statistics.

The twelve years of NOAA GML CO profiles from 2006–2017 provided the opportunity to
investigate the retrieval performance over time as shown in the AIRS and aircraft time series plot
of Fig. 14. There is a distinct seasonal cycle in the NOAA GML observations with high values
occurring during the northern hemisphere winter and lower values in the summer, which is also
captured by the AIRS retrievals. The bias drifts over this time period (Fig. 15) are small, $< 0.5\%$
per year in magnitude, for all levels and the column average. They are also of approximately the
same magnitude as those reported by Deeter et al. (2019) for MOPITT. There is a distinctive
seasonal cycle to the bias errors in middle and lower troposphere and column averages with biases
as high as 20% in the summer months and biases approaching zero during the winter months. We
hypothesize that this pattern is a result of greater photolytic destruction of the CO in the summer
months leading to lower background values not always captured by the retrieval perhaps due to
average a priori profiles being too high. We also examined the relationship between retrieval bias
and the CO mixing ratio. The bias sensitivity is the greater in the lower troposphere with average
biases at the 749 hPa pressure level ranging from positive 20% at low CO mixing ratios to near
zero at higher mixing ratios with an average slope of -0.16% per ppbv. At other levels and for the
column averages there is no marked dependence.


**5. Discussion and Conclusions**

A total of 15,112 quality-controlled AIRS single footprint CO retrievals were evaluated with a
total of 1,310 aircraft profiles from the HIPPO and ATom aircraft campaigns and the ongoing
NOAA GML measurement program. The retrievals were produced using the MUSES optimal
estimation algorithm that utilizes techniques first applied to the Aura TES instrument. The AIRS
profiles were matched with aircraft profiles with space and time coincidence criteria of 50 km and
9 hours. The aircraft profiles of CO mixing ratio were first convolved with the AIRS averaging
kernel to account for AIRS vertical sensitivity and then compared with the retrieved profiles. In



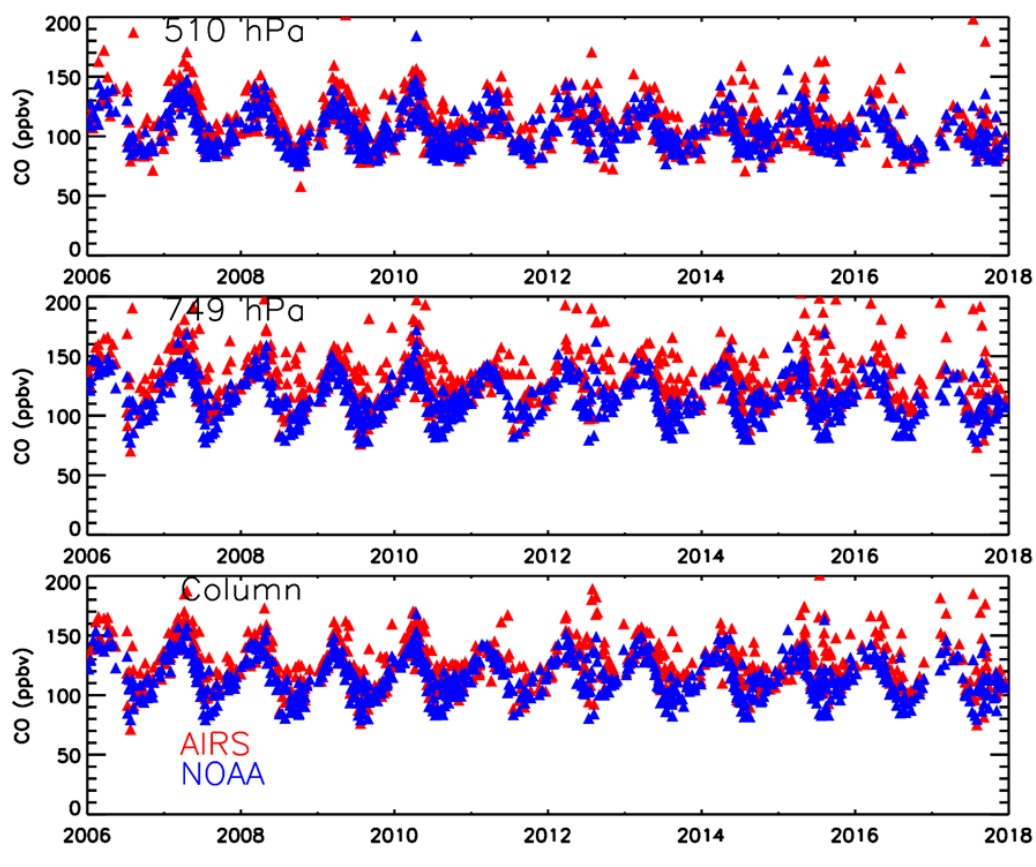


**Figure 14: AIRS MUSES CO retrieval (red) and corresponding NOAA GML observations (blue) for select pressure levels and the aircraft column averages.**






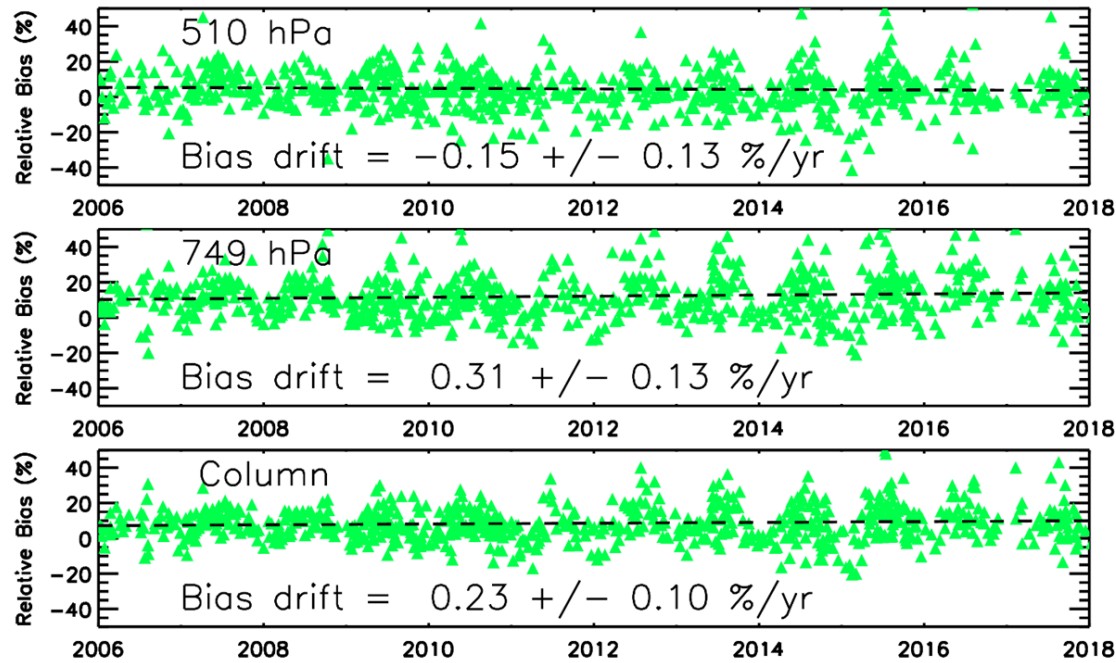

**Figure 15: AIRS MUSES CO retrieval relative bias (%) drift for select pressure levels and the aircraft column averages for the NOAA GML observations.**




addition, partial column average CO mixing ratios (referred to as column average mixing ratios for simplicity) defined as those between the highest and lowest aircraft flight level for each profile were estimated and compared to the corresponding AIRS values.

The averaging kernels generated by the MUSES algorithm indicated that the level of greatest AIRS sensitivity to CO was in the middle troposphere at or near the 510 hPa retrieval level. The estimated observation error also showed the lowest values at this level. Overall mean biases were +6.6% +/- 4.6%, +0.6% +/- 3.2%, -6.1% +/- 3.0%, and 1.4% +/- 3.6%, for 750 hPa, 510 hPa, 287 hPa, and the full column, respectively (Table 4). The mean standard deviations were 15%, 11%,

12%, and 9% at these same pressure levels, respectively.

For the HIPPO and ATom profile sets, the overall biases at the 510 hPa level were 0.95% and -1.10% respectively. For both HIPPO and ATom the AIRS CO comparison statistics had little sensitivity to land / ocean or day / night categorization. Column average mixing ratios by latitude for both sets exhibited lower mixing ratios in the 30S–90S band of about 50–70 ppbv with

increasing values toward the north reaching ~125–150 ppbv at 30N. While the column average errors were similar in both sets, the errors were highly skewed in the positive for HIPPO particularly in the 30N–60N latitude bands. Estimated theoretical observation errors from the AIRS MUSES algorithm indicated that the retrieval quality was generally within expected limits. However for HIPPO in the 30N–60N band the retrieval error standard deviation was ~4% higher

than expected and that was possibly due to the fact that the algorithm assumes a Gaussian error distribution and the errors were highly skewed in the positive in that region. The AIRS retrievals were able to distinguish between plume and background cases in the HIPPO case but were not always able to capture sharp vertical gradients or pinpoint the vertical location of the plume feature. The retrieval errors for the NOAA GML profiles were considerably higher than those for the

HIPPO and ATom sets. The 510 hPa and column average biases were 6.7% and 9.4% respectively. Like HIPPO, the column average errors were highly skewed in the positive suggesting a non-Gaussian distribution of errors and possibly explaining the much higher error standard deviation than the estimated theoretical observation error. The statistics of AIRS-aircraft differences were shown to be very sensitive to the values used to fill the aircraft profiles above the flight level due

to the propagation of error uncertainty to lower retrieval levels through the averaging kernel convolution procedure. Using a scaled a priori for the fill value resulted in a considerably smaller bias at the 510 hPa level of 0.7% and a slightly smaller column average bias of 5.8 %.



**Table 4: Summary statistics for all aircraft campaigns and categorizations.**

| | Bias 749.89 hPa (%) | STD 749.89 hPa (%) | Bias 510.90 hPa (%) | STD 510.90 hPa (%) | Bias 287.30 hPa (%) | STD 287.30 hPa (%) | Bias Column (%) | STD Column (%) |
|---|---|---|---|---|---|---|---|---|
| **All Average** | 6.6 | 14.7 | 0.6 | 11.0 | -6.1 | 12.4 | 1.4 | 8.9 |
| **All Standard Deviation** | 4.6 | 5.0 | 3.2 | 2.3 | 3.0 | 3.3 | 3.6 | 1.9 |


The results of the NOAA GML comparisons were more strongly affected by the choice of fill value above the flight level than the HIPPO or ATom comparisons since the NOAA GML profiles had a lower top with an average of 440 hPa compare to HIPPO and ATom with an average top at 290 hPa.

The twelve years of NOAA GML CO profiles from 2006–2017 provided the opportunity to evaluate the AIRS MUSES retrieval performance over time. The AIRS MUSES retrievals mostly capture the distinct observed seasonal cycle that featured higher mixing ratios in the winter and lower mixing ratios in the summer. However, the AIRS CO mixing ratios seemed to be biased high by ~20% in the summer in the lower troposphere. The bias drift for 2006 to 2017 was also

evaluated using the NOAA GML set and shown to be small (< 0.5 % per year).

**Data availability:**   The original HIPPO data file can be obtained from https://data.eol.ucar.edu/dataset/112.123. The NOAA GML data were obtained on request through Colm Sweeney through the NOAA GML Carbon Cycle Greenhouse Gases (CCGG) data

program. The ATom aircraft data were obtained from https://doi.org/10.3334/ORNLDAAC/1581 (Wofsy et al., 2018). AIRS-MUSES CO products are available via the GES-DISC from the NASA Tropospheric Ozone and Precursors from Earth System Sounding (TROPESS) project at https://disc.gsfc.nasa.gov/datasets/TRPSDL2COAIRSFS_1/summary.   The AIRS – aircraft matched dataset used here for validation is available from the authors on request.


**Author contributions:** JDH, VHP, KCP, SSK and JRW are responsible for the study design, data analysis, and manuscript writing. KCP was responsible for generating the AIRS MUSES retrievals. VK was responsible for managing the implementation of the MUSES retrieval algorithm software.





JRW contributed to the interpretation of validation results. HMW contributed to manuscript

editing. JVM, RC, BCD, EAK and KM were involved in making the HIPPO, ATom and NOAA GML aircraft measurements and provided guidance on the use of these measurements in the validation process.

**Competing interests:** The authors declare that they have no conflict of interest.


**Acknowledgements:** Part of this research was carried out at the Jet Propulsion Laboratory, California Institute of Technology, under a contract with NASA. Part of the research was carried out at Atmospheric and Environmental Research Inc. with funding from the JPL TROPESS team. The NOAA GML aircraft observations are supported by NOAA. The HIPPO aircraft data were
supported by NOAA and NSF. The ATom aircraft data were supported by NASA.

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
