# Peer review of "Validation and Error Estimation of AIRS MUSES CO Profiles with HIPPO, ATom and NOAA GML Aircraft Observations"

_Atmospheric Measurement Techniques, 2021_

## Referee Comment (RC2)

The authors evaluated AIRS MUSES CO profiles and retrieval error estimates with HIPPO, ATom, and NOAA GML aircraft observations. The following comments need to be addressed.

Figure 1: Please also add lines in addition to the dots so that the flight tracks are clearer.

Line 183: Please change "Thus, each aircraft profile was evaluated against a set of AIRS profiles." to "Thus, each aircraft profile was compared to a set of AIRS profiles".

Line 185: Could you specify how many levels are there in the AIRS MUSES forward model?

Figures 3 and 7: "and the latitude bands are indicated in the upper left." The latitude bands are missing in the Figures. And the sentence "The red lines indicate the individual profiles, the black solid lines the mean difference or bias, and the dashed lines one standard deviation from the mean." needs to be corrected.

Figure 4: Please add a legend to figure 4b, since you have a legend for black dots in figure 4a, and black dots represent different things in 4a and 4b.

Section 4: I'm a little confused with the comparisons between "AIRS-aircraft standard deviation" and "a priori–aircraft standard deviation" (Figures 5, 9, 13). Could you explain more on what is "AIRS-aircraft standard deviation" and "a priori–aircraft standard deviation"? If "AIRS-aircraft standard deviation" stands for "standard deviation of the difference between a priori and aircraft", then this value only represent the variability of the bias of a priori from aircraft instead of magnitude of the bias. Therefore when AIRS-aircraft standard deviation is lower than a priori–aircraft standard deviation, it only means that the bias of AIRS has smaller variability than the bias of a priori. How could this indicate that AIRS perform better than a priori? Am I understanding it right? Do you mean "the square root mean of the difference between a priori and aircraft"?

Line 310: "......variability within the set of AIRS profiles collocated with an aircraft profile, which can be thought of as an empirical error." To me the variability within the set of AIRS profiles collocated with an aircraft profile is representativeness error. I'm wondering if it is the same as what's discussed here and is it comparable to the theoretical error? As also shown by Figures 6 and 10, comparisons to plume obs show higher empirical error values than comparisons to background obs. This is because representativeness error is higher in plume (more heterogeneous) compared to background (less heterogeneous), and is not related to the theoretical errors. The comparisons to variability within the set of AIRS profiles collocated with an aircraft profile do not seem necessary to the main story of the manuscript. However, if the authors do include this part of the comparisons, please provide overall statistics of the empirical errors and theoretical errors in addition to the illustrative cases. And please also discuss what does it mean when the empirical errors and theoretical errors are close or far away.

Figure 8 are different from Figure 4. In Figure 8b, the average differences are positive in -30S-10N band. And the average difference is negative at 30N, which is opposite to Figure 4b. Please add a brief discussion for this.

And I was also wondering if the same a prior profiles were used for ATOM and HIPPO periods?

Because the mean a priori error estimate for ATOM (Figure 9) is higher than that for HIPPO (Figure 5), which may partially contribute to the "better" retrieval performance relative to the prior for the ATom vs the HIPPO comparisons.

Line 349: "The distribution of errors in the 30N–60N latitude band is less skewed than for HIPPO (0.54 vs. 1.36) suggesting that a Gaussian distribution of errors is a reasonable assumption for this dataset." But outside the 30N-60N, distribution of errors seems more skewed for this dataset (e.g., -30S-10N).

Line 560: "We also examined the relationship between retrieval bias and the CO mixing ratio." Please add a figure for this if possible.

---

## Author Comment (AC1)

*Response to RC1:*

*Thank you for your thorough review and thoughtful comments on our manuscript. We agree that we did not adequately explain the significance of our validation results for monitoring Earth system change. We have revised the manuscript to address this important concern. Our responses to each of your comments and questions are listed below in italic font.*

**RC1**: 'Comment on amt-2021-211', Anonymous Referee #1, 23 Aug 2021   reply
The authors present an in-depth error analysis of the AIRS MUSES CO retrieval product. The MUSES algorithm follows the Rodgers optimal estimation approach (OE), which allows them to quantify a smoothing error, measurement noise, systematic uncertainty, cross-state error and retrieval residual. They use these to derive four error terms – theoretical, a-priori, retrieval, and empirical – for MUSES CO retrievals collocated with aircraft observations made during the HIPPO, ATom and NOAA GML campaigns. For each campaign, the authors repeat their analysis and present four figures and a table.

This paper summarizes an extraordinary amount of meticulous work about an important topic in satellite retrieval theory and application today, namely error quantification and validation. The authors acknowledge the importance of understanding all the retrieval error terms if we are to use a satellite record in climate science and the quantification of Earth system change. This is especially true for the AIRS record that now spans two decades. But as this paper makes clear, this is no easy task. I have spent some time with this paper, revisiting sections, and am left with the conclusion that while the authors present clear, detailed results, they fail to communicate what it all means. It would have been very helpful to the reader if the authors explained how these results will be used to improve the MUSES algorithm (or product) going forward, or what this error evaluation means for the AIRS record and its application in Earth system science. After working through all the details, trying to understand the results, I (the reader) am left thinking "so what?". This paper can make a meaningful, even important, contribution if the authors elaborate on the significance of their results in the Discussion/Conclusion.

*This validation study and the results presented in this paper are significant for the following reasons.*

*1. We validate AIRS single footprint retrievals which can better resolve smaller pollution plumes, such as those generated by small-scale fires than the operational AIRS product that is produced from the 3 x3 array of AIRS footprints. For example, the plot below shows single footprint AIRS CO retrievals from the WE-CAN campaign as colored circles with the bounding box of the 3 x 3 array field of regard overlaid. The shape of the CO plumes and the higher values from the fires are well represented but would be smoothed out in the lower-resolution AIRS operational product. While this figure is not part of our study and not included in the paper it illustrates visually the impact of finer spatialresolution.*

[Figure]

To better emphasize the importance of the higher resolution we added the following sentences to the Introduction starting on line 89 of the original manuscript.
"The improved spatial resolution enables better representation of smaller pollution plumes from local strong anthropogenic sources and small wildfires which will enable better pollution tracking and more precise trend analysis.  For example, George et al ( 2009) found that CO related to fires was systematically ~ 17 % lower for AIRS than MOPITT and IASI due to AIRS's coarser resolution. Furthermore, Buchholz et al (2021) using MOPIIT found that recent trends in column CO over northeastern China were driven mainly by significant trends in the 75th  percentile values suggesting changes in local rather than regional emission sources."

2. The validation of the observation error (as reported in the satellite products), spatial variation in bias and any time drift in the bias are important for use of this AIRS CO data in emissions estimation and/or chemical reanalysis.  Our results do not show an appreciable latitudinal dependence in the bias. Furthermore, our results show that the bias drift over time is small (~0.5%).

3. Our results suggest that the reported /estimated  observation errors are low by a factor of up to ~2. In Figure 5 ( particularly in the 30N – 60N band)  and Figure 13  the AIRS- Aircraft standard deviations are much larger than the reported observation errors.  This indicates that observation errors are an underestimate of  the actual retrieval error.  The cause is not yet determined but is being investigated.

We have made substantial revisions to the Discussions and Conclusions section to better emphasize points 1-3 above.

Review:

- Lines 37-39: "We find mean biases of + 6.6% +/- 4.6%, +0.6% +/- 3.2%, -6.1% +/- 3.0%, and 1.4% +/- 3.6%, for 750 hPa, 510 hPa, 287 hPa, and the column averages, respectively. The mean standard deviation is 15%, 11%, 12%, and 9% at these same pressure". This sentence is very difficult to read and I suggest rephrasing it to help clarify one of the main results of this paper.

  *These sentences have been changed to the following. "We found mean biases of + 6.6% +/- 4.6%, +0.6% +/- 3.2%, and -6.1% +/- 3.0% for three representative pressure levels of 750 hPa, 510 hPa, 287 hPa, and column average mean biases of 1.4% +/- 3.6%. The mean standard deviations for the three representative pressure levels were 15%, 11% and 12% and the column average standard deviation was 9%."*

- Line 143: "Atmospheric Infrared Sounder (AIRS)" already defined

  *Changed "The Atmospheric Infrared Sounder (AIRS) is" to "AIRS is".*

- Line 164: "CO is retrieved using the 2181-2200 cm-1 spectral range". Does MUSES use all channels in this spectral range for its CO retrieval?

  *Yes, this is correct.*

- What does MUSES use as a-priori for CO? I think it is important to state this clearly in the paper given the results presented later.

  *The a priori profiles for CO are derived from from a monthly climatology, in 30 degree latitude by 60 deg longitude boxes and there is no variation in this climatology from one year to the next. The climatology was produced from MOZART model output, from runs performed for construction of climatologies for the Aura mission [Brasseur et al., 1998]. The a priori constraint used for CO is the same constraint used in the MOPITT CO algorithm [Deeter et al., 2010].*

  *the MOZART model. We have added the following to Line 225 of the original manuscript.*

  *"For AIRS MUSES CO retrievals, the a priori profiles are obtained from a monthly climatology, in 30 degree latitude by 60 deg longitude boxes produced from the MOZART atmosphere chemistry model for the Aura mission (Brasseur et al., 1998). The a priori constraint used for CO is the same constraint used in the MOPITT CO algorithm [Deeter et al., 2010]."*

  *"Brasseur, G. P., Hauglustaine, D. A., Walters, S., Rasch, P. J., Muller, J. F., Granier, C., and Tie, X. X.: MOZART, a global chemical transport model for ozone and related chemical tracers 1. Model description, J. Geophys. Res.-Atmos., 103, 28265–28289, 1998."*

*Deeter, M. N., et al. (2010), The MOPITT version 4 CO product: Algorithm enhancements, validation, and long-term stability, J. Geophys. Res., 115, D07306, doi:10.1029/2009JD013005*

*We have also added the citations above to the References section.*

- MUSES does not follow the AIRS Science Team approach of deriving an aggregate clear-sky radiance from each 3 x 3 array of AIRS measurements in partly cloudy skies. While cloud clearing reduces the spatial resolution of the radiance measurements ahead of retrieval, it not only allows stable retrievals in complex scenes but also allows the quantification of uncertainty due to clouds (Smith and Barnet, 2019; Susskind et al., 2014; Maddy et al., 2009; Chahine, 1982, 1977). Clouds being one of the primary sources of scene-dependent uncertainty, this is an important source to account for in an error analysis. How does MUSES quantify systematic uncertainty due to clouds? And given the results presented, can the authors draw any conclusions about the validity of retrieving CO from AIRS measurements in the presence of clouds?

*MUSES retrieves cloud optical depth following Kulawik et al ( 2006). The retrieval provides both spectrally dependent and full spectrum average effective cloud OD and an effective OD. In our analysis the effective cloud optical depths to remove retrievals using a threshold effective cloud optical depth of 0.1. We have added the following text starting on Line 178 of the original manuscript.*

*"While the AIRS MUSES algorithm uses the original single pixel instrument radiances rather than cloud-cleared radiances, the algorithm does retrieve cloud optical thickness following Kulawik et al (2006) and provides both a spectrally varying and average effective optical depth. The cloud optical depth is retrieved before CO, thus the effect of clouds is taken into account in the CO retrieval. AIRS MUSES profiles with optically thick clouds were designated as those with an average cloud effective optical depth over the AIRS spectrum and within the CO absorption band greater than 0.1 and were removed from the set."*

*We have also added to the Reference section the following citation.*

*"Kulawik, S. S., Worden, J., Eldering, A., Bowman, K., Gunson, M., Osterman, G. B., Zhang, L., Clough, S. A., Shephard, M. W., and Beer, R.: Implementation of cloud retrievals for Tropospheric Emission Spectrometer (TES) atmospheric retrievals: part 1. Description and characterization of errors on trace gas retrievals, J. Geophys. Res.-Atmos., 111, D24204, https://doi.org/10.1029/2005JD006733, 2006."*

- Line 175: "the original non cloud-cleared radiances". It will be more straightforward to simply say "the instrument radiances"

*We think it is important to emphasize here that the radiances used have not been subjected to cloud clearing and therefore could be affected by the presence of optically thick clouds. That sentence has been changed to read "Since the AIRS MUSES algorithm uses the original instrument radiances rather than cloud-cleared*

*radiances, profiles with optically thick clouds diagnosed by the AIRS MUSES algorithm were also removed from the set."*

- Line 178: "profiles with thick clouds were also removed from the set." How did the authors distinguish thick (versus thin) clouds?

  *Thick clouds would be those with an effective cloud optical depth greater than 0.1. To make this point clearer the sentence starting on Line 178 has be rewritten as "AIRS MUSES profiles with optically thick clouds were designated as those with an average cloud effective optical depth over the AIRS spectrum and within the CO absorption band greater than 0.1."*

- Figure 2 needs to be resized.

  *We can resize it (make it smaller) in the final draft.*

- It is difficult to make sense of the results in Tables 1 through 3. I think a single figure summarizing the values from all three would have made it easier to inter-compare among latitude zones and aircraft campaigns.

  *Table 4 shows average or summary statistics for all campaigns. We have also added three rows for the overall statistics for each of the three campaigns to this table to facilitate inter-campaign comaprisons.*

- Lines 285-286: "Beyond examining biases and variability of the retrieved profiles, evaluating the retrieval error estimates is also important, since they provide users with a measure of the reliability of the data". I agree with this statement in principal but "provid[ing] users with a measure of the reliability of the data" is what bias and standard deviation tells the user at first order. What, in the authors experience, do an error analysis contribute over and above a measure of reliability? Perhaps the authors can clarify this point with examples on how such an error analysis influences algorithm design/updates and data application.

  *As stated in our response to the general comments our results suggest that the estimated observation errors are low by a factor of up to ~2, particularly in the 30N – 60N band. Furthermore, the AIRS- Aircraft standard deviations are much larger than the reported observation errors. These findings indicate that observation errors are an underestimate of the actual retrieval error. The cause of this underestimate is still under investigation.*

- Figures 5, 9, 13:

Is it correct to interpret this figure as meaning that the MUSES retrieval basically added noise to the a-priori between the Earth surface and 600 hPa?

*No, this just indicates where the sensitivity is and is dependent upon natural variability in the retrieved scenes (e.g., T, Q, CO etc.) and large spatiotemporal coincidence criteria.*

How sensitive are these error values to variation in a-priori error?

*The observation errors shown in Figs 5, 9, 13 are not sensitive to variation in the a priori error covariance. The observation error is the sum of the covariances associated with noise and cross-state terms. The smoothing error term (which is impacted by the a priori covariance) is not included in the observation error.*

What would be an ideal relationships between all these error terms?

*Ideally you would be measuring exactly the same air mass and the computed observational error would match the standard deviation of AIRS – Aircraft. However, this is not feasible, since AIRS will always measure a larger volume vertically and horizontally than the aircraft, even if the latter is spiraling.*

Do the authors think that the observation error in the lower troposphere will change if they adjust the a-priori error vertically?

*It would change the profile retrieval and smoothing error but not observational error.*

The legend states "mean observation error", "mean a priori error", 'AIRS-AIRCRAFT Std. Dev" and "A Priori-AIRCRAFT Std. Dev", but in the text associated with these figures, the authors discuss the "theoretical error" and "empirical error". It will help the reader a great deal if the authors maintain consistency in their terminology.

Please note that the references to "mean observation error", "mean a priori error", 'AIRS-AIRCRAFT Std. Dev" and "A Priori-AIRCRAFT Std. Dev" in the text are referring to Figures, 5, 9 and 13 while the terms "theoretical error" and "empirical error" refer to an alternative error analysis approach illustrated in Figures 6 and 10. However, it is correct that the "theoretical error" in Figures 6 and 10 is the calculated in the same way as the "mean observation error" but just for a set of AIRS profiles collocated with a single aircraft profile rather than an entire latitude band.

*We have changed the legends in Figures 6 and 10 and the text referring to them to maintain consistency. The term "theoretical error" has been replaced with "mean observation error" ( to be consistent with the terminology of Figures 5, 9 and 13) and the term "empirical error" has been replaced with "AIRS profile variability", which is a more descriptive name. Note that the "AIRS profile variability" was estimated as the square root of the diagonal of the covariance matrix of all the coincident AIRS MUSES retrievals with a particular aircraft location and is different from the AIRS – AIRCRAFT standard deviation which is a measure of the variability of the actual retrieval error.*

*We have also changed the introductory information related to the alternative approach related to Figures 6 and 10 on Lines 310 – 312 in the original manuscript to the following.*

*"An alternative approach for evaluating the theoretical error is to compare it to the variability within the set of AIRS profiles collocated with an aircraft profile.   If it is assumed that all satellite footprints in the collocated set are basically seeing the same scene then the*

*variability in the retrieved profiles can be considered an empirical error (Oetjen et al., 2014).*
*In this analysis this empirical error is referred to simply as the AIRS profile variability."*

Do these results mean that an end user should use the a-priori instead of the retrieval in the
lower troposphere?

*The retrieval products include the averaging kernel for each retrieved profile. The averaging*
*kernels are scene-dependent. For each profile, the averaging kernel provides information*
*on how strongly the retrieved profile is influenced by the prior vs the measured radiances at*
*any given altitude. The end user should use the retrieved profile together with the averaging*
*kernel.*

*In general, the AIRS CO retrievals do not provide information on CO variations close to the*
*surface, so if the end user is primarily interested in near-surface CO, they should be looking*
*for a different product.*

- Line 277/Line 355: Figure 3/Figure 7 captions, "The number of profiles and the
  latitude bands are indicated in the upper left". I only see the sampling size listed, not
  the latitude band.

  *We have added the latitude band information to these figures.*

- Figure 11 should be resized.

  *We will resize this figure for the final draft.*

- Line 603: "AIRS MUSES algorithm indicated that the retrieval quality was generally
  within expected limits." What are these limits?

  *We have changed this statement to read "Estimated  theoretical observation errors*
  *from the AIRS MUSES algorithm were generally small as expected in the middle*
  *troposphere where AIRS has good sensitivity."*

- Given this in-depth analysis of multiple error sources, what does it mean for AIRS
  product design? What are the lessons learned for algorithm teams and/or end user
  applications?

  *The important findings for the end users are that there doesn't appear to be a strong*
  *latitudinal dependence of bias and that the bias drift over time is small.  This*
  *suggests that the data can be used to compare regional differences in CO mixing*
  *ratios and to track trends over time.*

  *For the algorithm team the fact that the observation errors are underestimating the*
  *actual retrieval errors suggests that adjustments could be made to the constraints*
  *used in the algorithm.*

References

Chahine, M. T.: Remote sounding of cloudy parameters.II. Multiple cloud formations, J. Atmos. Sci., 34, 744–757, 1977.

Chahine, M. T.: Remote sensing of cloud parameters, J. Atmos. Sci., 39, 159–170, 1982.

Maddy, E. S., Barnet, C. D., and Gambacorta, A.: A computationally efficient retrieval algorithm for hyperspectral sounders incorporating a-priori information, 6, 802–806, https://doi.org/10.1109/LGRS.2009.2025780, 2009.

Smith, N. and Barnet, C. D.: Uncertainty Characterization and Propagation in the Community Long-Term Infrared Microwave Combined Atmospheric Product System (CLIMCAPS), 11, 1227, https://doi.org/10.3390/rs11101227, 2019.

Susskind, J., Blaisdell, J. M., and Iredell, L.: Improved methodology for surface and atmospheric soundings, error estimates, and quality control procedures: the atmospheric infrared sounder science team version-6 retrieval algorithm, 8, 084994, https://doi.org/10.1117/1.JRS.8.084994, 2014.

---

## Author Comment (AC2)

*Response to RC2:*

*Thank you for your thorough review and thoughtful comments on our manuscript. Our responses to each of your comments and questions are listed below in italic font*

The authors evaluated AIRS MUSES CO profiles and retrieval error estimates with HIPPO, ATom, and NOAA GML aircraft observations. The following comments need to be addressed.

Figure 1: Please also add lines in addition to the dots so that the flight tracks are clearer.

*It is more correct to say that Figure 1 shows the positions of all the aircraft profiles used in this study rather than the actual flight tracks. There are gaps where no usable aircraft and/or coincident satellite data were available. At Line 118 in the original manuscript, we have added the following sentence.*

*"The locations of all the aircraft profiles used in this study are shown in Fig. 1."*

*The first sentence of the figure caption for Fig. 1 has also been revised to "Locations of aircraft profiles used for HIPPO and ATom as colored dots and NOAA GML as black diamonds with 3-character string identifier. "*

Line 183: Please change "Thus, each aircraft profile was evaluated against a set of AIRS profiles." to "Thus, each aircraft profile was compared to a set of AIRS profiles".

*This line has been changed*

Line 185: Could you specify how many levels are there in the AIRS MUSES forward model?

*There are 67 levels, and that information has been added to the text.*

Figures 3 and 7: "and the latitude bands are indicated in the upper left." The latitude bands are missing in the Figures. And the sentence "The red lines indicate the individual profiles, the black solid lines the mean difference or bias, and the dashed lines one standard deviation from the mean." needs to be corrected.

*The latitude bands have been added to these figures.*

Figure 4: Please add a legend to figure 4b, since you have a legend for black dots in figure 4a, and black dots represent different things in 4a and 4b.

*Legends have been added to the bottom panels of Figures 4, 8 and 12 indicating that the blue triangles are the AIRS -AIRCRAFT differences and the black dots are the AVERAGE AIRS - AIRCRAFT differences in the 10-degree bins.*

Section 4: I'm a little confused with the comparisons between "AIRS-aircraft standard deviation"

and "a priori–aircraft standard deviation" (Figures 5, 9, 13). Could you explain more on what is "AIRS-aircraft standard deviation" and "a priori–aircraft standard deviation"? If "AIRS-aircraft standard deviation" stands for "standard deviation of the difference between a priori and aircraft", then this value only represent the variability of the bias of a priori from aircraft instead of magnitude of the bias. Therefore when AIRS-aircraft standard deviation is lower than a priori–aircraft standard deviation, it only means that the bias of AIRS has smaller variability than the bias of a priori. How could this indicate that AIRS perform better than a priori? Am I understanding it right? Do you mean "the square root mean of the difference between a priori and aircraft"?

*For these large datasets RMS and standard deviation are equivalent. When the AIRS -aircraft standard deviation is smaller than the a priori – aircraft standard deviation it indicates that the retrieval is doing a better job at capturing the actual variability in the aircraft profiles than the a priori. Furthermore, when the AIRS – aircraft standard deviation is greater than the mean observation error it indicates that the observation error is underestimating the actual retrieval error.*

Line 310: "……variability within the set of AIRS profiles collocated with an aircraft profile, which can be thought of as an empirical error." To me the variability within the set of AIRS profiles collocated with an aircraft profile is representativeness error. I'm wondering if it is the same as what's discussed here and is it comparable to the theoretical error? As also shown by Figures 6 and 10, comparisons to plume obs show higher empirical error values than comparisons to background obs. This is because representativeness error is higher in plume (more heterogeneous) compared to background (less heterogeneous), and is not related to the theoretical errors. The comparisons to variability within the set of AIRS profiles collocated with an aircraft profile do not seem necessary to the main story of the manuscript. However, if the authors do include this part of the comparisons, please provide overall statistics of the empirical errors and theoretical errors in addition to the illustrative cases. And please also discuss what does it mean when the empirical errors and theoretical errors are close or far away.

*The terms theoretical error and empirical error were adopted from Oetjen et al (2014) who performed a similar analysis on satellite and ozonesonde data. In their analysis Oetjen et al (2014) assumed that the coincident satellite profiles are basically seeing the same scene and therefore any variability in the retrieved profiles could be considered an empirical measure of the retrieval error. Yes, it is likely that the plume cases would probably feature more actual heterogeneous mixing ratios compared to the background cases and therefore empirical errors are more likely to be much greater from a statistical basis. However, our illustrative cases were selected to show the range of retrieval performance. For example, in Figure 6, the empirical errors for the HIPPO2 plume case are very large and much larger than the theoretical errors, but for the HIPPO3 case they are much smaller and similar to the theoretical errors.*

*For clarity and consistency, we have changed the names of the terms used for this analysis in the revised manuscript. The theoretical error is now called the mean observation error to be consistent with the terminology used in Figures, 5, 9 and 13. The empirical error has been renamed the AIRS profile variability.*

*The introductory information for this approach on Lines 310 – 312 of the original manuscript has been changed to the following.*

*"An alternative approach for evaluating the theoretical error is to compare it to the variability within the set of AIRS profiles collocated with an aircraft profile. If it is assumed that all satellite footprints in the collocated set are basically seeing the same scene then the variability in the retrieved profiles can be considered an empirical error (Oetjen et al., 2014). In this analysis this empirical error is referred to simply as the AIRS profile variability."*

Figure 8 are different from Figure 4. In Figure 8b, the average differences are positive in -30S-10N band. And the average difference is negative at 30N, which is opposite to Figure 4b. Please add a brief discussion for this. And I was also wondering if the same a prior profiles were used for ATOM and HIPPO periods? Because the mean a priori error estimate for ATOM (Figure 9) is higher than that for HIPPO (Figure 5), which may partially contribute to the "better" retrieval performance relative to the prior for the ATom vs the HIPPO comparisons.

*The HIPPO and ATom campaigns were conducted in different years and it appears that there were differences in the CO mixing ratios in the air masses sampled during the respective campaigns. For HIPPO the aircraft column average CO mixing ratios in the 30S – 10N band were all less than 100 ppbv, whereas for ATom they were much more variable and were as high as ~ 130 ppbv. Between 30N – 40N the HIPPO column mixing ratios ranged from ~70 ppbv to ~ 140 ppbv whereas for ATom they were lower ranging from ~60 ppbv to ~125 ppbv.*

*The a priori profiles used for the HIPPO and ATom retrievals were obtained from a climatology generated with the MOZART atmospheric chemistry model(Brasseur et al., 1998). A reference to this model has been added to the revised manuscript in Section 3.2. The differences in the a priori errors are due to the differences in the air masses (with different CO mixing ratios) sampled during the different years of the campaigns.*

*Brasseur, G. P., Hauglustaine, D. A., Walters, S., Rasch, P. J., Muller, J. F., Granier, C., and Tie, X. X.: MOZART, a global chemical transport model for ozone and related chemical tracers 1. Model description, J. Geophys. Res.-Atmos., 103, 28265–28289, 1998.*

Line 349: "The distribution of errors in the 30N–60N latitude band is less skewed than for HIPPO (0.54 vs. 1.36) suggesting that a Gaussian distribution of errors is a reasonable assumption for this dataset." But outside the 30N-60N, distribution of errors seems more skewed for this dataset (e.g., -30S-10N).

*We looked at the skew only in the latitude bands corresponding to those in Figures 5, 9 and 13 and found that the significant positive skew in the 30N -60N band for HIPPO was a plausible explanation for the failure of the diagnosed observation errors to represent actual retrieval errors in those profiles.*

Line 560: "We also examined the relationship between retrieval bias and the CO mixing ratio." Please add a figure for this if possible.

*We have added this figure as Fig. 16 and included and added the reference "(Fig. 16)" to the end of the sentence above.*